# Systematically and efficiently improving $k$-means initialization by pairwise-nearest-neighbor smoothing

**Carlo Baldassi**                                                          *carlo.baldassi@unibocconi.it*
*Department of Computing Sciences, ArtLab, BIDSA*
*Bocconi University, Milan*
*ELLIS Scholar*

**Reviewed on OpenReview:** *https://openreview.net/forum?id=FTtFAg3pek*

## Abstract

We present a meta-method for initializing (seeding) the $k$-means clustering algorithm called PNN-smoothing. It consists in splitting a given dataset into $J$ random subsets, clustering each of them individually, and merging the resulting clusterings with the pairwise-nearest-neighbor (PNN) method. It is a meta-method in the sense that when clustering the individual subsets any seeding algorithm can be used. If the computational complexity of that seeding algorithm is linear in the size of the data $N$ and the number of clusters $k$, PNN-smoothing is also almost linear with an appropriate choice of $J$, and quite competitive in practice. We show empirically, using several existing seeding methods and testing on several synthetic and real datasets, that this procedure results in systematically better costs. In particular, our method of enhancing $k$-means++ seeding proves superior in both effectiveness and speed compared to the popular "greedy" $k$-means++ variant. Our implementation is publicly available at https://github.com/carlobaldassi/KMeansPNNSmoothing.jl.

## 1 Introduction

The classical $k$-means algorithm is one of the most well-known and widely adopted clustering algorithms (Berkhin, 2006; Wu et al., 2008). Given $N$ data points $\mathcal{X} = (x_i)_{i=1..N}$, where each point is $D$-dimensional, $x_i \in \mathbb{R}^D$, and given an integer $k \geq 2$, the algorithm aims at minimizing the sum-of-squared-errors (SSE) cost, defined as a function of $k$ centroids $\mathcal{C} = (c_a)_{a=1..k} \in (\mathbb{R}^D)^k$, and of a partition of the data points $\mathcal{P} = (p_i)_{i=1..N} \in \{1, \ldots, k\}^N$, as such:

$$\text{SSE}(\mathcal{C}, \mathcal{P}; \mathcal{X}) = \sum_{i=1}^N \|x_i - c_{p_i}\|^2 \tag{1}$$

For fixed $\mathcal{C}$, the optimal partition is obtained by associating each point to its nearest centroid, and conversely, for fixed $\mathcal{P}$, the optimal centroids are given by the barycenter of each cluster. The $k$-means algorithm starts from an initial guess for the configuration and alternates optimizing $\mathcal{C}$ and $\mathcal{P}$ until a fixed point is reached. This alternating procedure, due to Lloyd (Lloyd, 1982), is greedy and converges to a local minimum. Its computational cost is $O(kND)$ per iteration; both optimization steps can be straightforwardly parallelized over $N$. It is notoriously sensitive to the choice of the initial configuration, i.e. the seeding, both in terms of the final value of the SSE and of the number of Lloyd's iterations required to converge. Several schemes have been proposed, with various degrees of complexity. An extremely basic and cheap option is to sample the initial centroids uniformly at random from $\mathcal{X}$ (MacQueen, 1967). Other popular methods typically produce considerable improvement and also scale like $O(kND)$, e.g. $k$-means++ (Arthur & Vassilvitskii, 2007) and maxmin (Gonzalez, 1985; Katsavounidis et al., 1994), among several others. Yet other methods have worse scalings and thus tend to dominate the computational time, e.g. the pariwise-nearest-neighbors (PNN) method (Equitz, 1989) which scales like $\Omega(N^2)$.

In this paper, we propose a novel scheme, PNN-smoothing, based on randomly splitting the dataset $\mathcal{X}$ into $J$ subsets, clustering them individually with $k$-means, and then merging the resulting $Jk$ clusters following the PNN procedure until only $k$ clusters remain: these constitute the new seed for Lloyd's algorithm. It is a meta-method, in the sense that it can use any seeding procedure for the subsets. We denote this with PNNS(INIT) where INIT is any seeding algorithm. If INIT is $O(kN)$, then PNNS(INIT) is also almost linear, as long as we set $J = O\left(\sqrt{N/k}\right)$. Our empirical tests indicate that PNNS(INIT) gives systematically better SSEs compared to INIT, at a very small computational cost, even in a parallel implementation. Its results are surprisingly good even when INIT is one of the worst seeding methods, random uniform initialization.

Throughout the paper, we focus exclusively on the effect of seeding on basic $k$-means, which can be regarded as a basic tool in the optimization of the SSE cost. Evaluating and contrasting the effectiveness of different methods is not straightforward. Our main metric will be the wall-clock computational time, measured using a state-of-the-art implementation of all methods under uniform conditions. It is also desirable, however, to provide some hardware- and language-independent metric, beyond the asymptotic scaling analysis. It is often the case that computing distances between vectors takes up the majority of the computational effort and thus that the number of distance computations correlates well with the running time, especially in high-dimensional cases (although of course various circumstances, especially caching, can significantly affect the computational cost of a distance computation in practice) (Newling & Fleuret, 2016). Indeed, several methods employed to accelerate the Lloyd's iteration procedure are explicitly aimed at reducing the number of distance computations (in the $\mathcal{P}$-from-$\mathcal{C}$ step) at the cost of some additional bookkeeping (Kaukoranta et al., 1999; Elkan, 2003; Hamerly, 2010; Ding et al., 2015; Newling & Fleuret, 2016; Xia et al., 2022).

We thus define the number of normalized distance computations (NDC) as the total number of distance (or squared-distance) computations between $D$-dimensional vectors, wherever they may appear in a procedure, divided by $Nk$. The normalization makes this measure comparable across datasets, so that a value of 1 corresponds to one computation of the partition $\mathcal{P}$ from $\mathcal{C}$, performed from scratch. Due to the variability in the optimization procedure, and the use of accelerators, it is generally quite difficult to estimate the NDC required by a given method. However, for some seeding algorithms the number of NDC can be computed exactly, and for others it is rather straightforward to at least provide some useful lower bounds. Note that our implementation of Lloyd's algorithm starts by optimizing $\mathcal{C}$ from $\mathcal{P}$, and thus we regard seeding algorithms as producing an initial partition $\mathcal{P}$. This choice is justified by the fact that several seeding algorithms, even if they are based on picking some initial centroids, include the computation of $\mathcal{P}$ as a byproduct, and those that don't would need to perform this step at least once in any case[1]; therefore, this definition allows to compare all seeding algorithms consistently. As a consequence, all the seeding algorithms that we will consider in this work require at least 1 NDC.

The rest of the paper is organized as follows. In sec. 2 we review prior literature and describe a few seeding methods that will be considered in the tests. In sec. 3 we describe and discuss in detail the PNNS(INIT) scheme. In sec. 4 we present and analyze detailed numerical results on several challenging synthetic and non-synthetic datasets. Sec. 5 has a final discussion.

## 2 Relation to prior works

As mentioned in the introduction, a large number of seeding schemes for $k$-means have been proposed. Extensive reviews and benchmarks can be found in refs. (Celebi et al., 2013; Fränti & Sieranoja, 2019). Here, we only cover a few selected ones, chosen on criteria of simplicity, popularity, similarity with our scheme, and effectiveness. Their scaling and NDCs characteristics (together with those of PNNS(INIT) described in the next section) are also summarized in table 1.

**UNIF.** Uniformly sampling (preferably without replacement) $k$ centroids from the dataset (MacQueen, 1967) is arguably the most popular method. We'll call this seeding method UNIF. It's extremely simple, but its performance is generally very poor, even in moderately hard circumstances, as it often leads to poor local minima and long convergence times.

---

[1] We only consider the exact Lloyd's iterations here, in which case sub-linear (in $N$) seeding methods such as AF-KMC[2] (Bachem et al., 2016) don't provide an advantage.

For the reasons explained in the introduction, although the cost of sampling the centroids is $O(k)$, the overall cost of UNIF is still $O(kND)$, entirely due to performing 1 NDC.

**MAXMIN.** Another simple method (Gonzalez, 1985; Katsavounidis et al., 1994) goes under the name of "furthest point", or "maxmin", or "maximin". We will refer to the same variant that was used in Celebi et al. (2013); Fränti & Sieranoja (2019), and call it MAXMIN. It consists in selecting the first centroid at random from the dataset, after which the process is iterative and deterministic: at each step, each successive centroid is chosen as the furthest point from the centroids selected so far. More precisely, it's the point that maximizes the distance from its nearest centroid: $c_a = \text{argmax}_{x \in \mathcal{X}} \min_{b<a} \|x - c_b\|^2$.

This algorithm scales as $O(kND)$; it also computes the optimal partition (with respect to the chosen centroids) as a byproduct of the selection procedure: it requires precisely 1 NDC. The results of Fränti & Sieranoja (2019), on synthetic datasets, report this method as being among the best of those that were tested. On the other hand, in Celebi et al. (2013), in which an array of real datasets was also tested, the authors advise against this method. They suggest instead, among the algorithms in the same complexity class, to use greedy-$k$-means++ or Bradley and Fayyad's "refine" (both described below).

**[G]KM++.** The "$k$-means++" seeding method (Arthur & Vassilvitskii, 2007) can be regarded as a more stochastic version of MAXMIN. While the first centroid is also selected at random from the dataset, the remaining centroids are sampled from the dataset with a probability proportional to the squared distance from the closest centroids. More precisely, the probability of selecting a point $x$ as the next centroid $c_a$ when $a \geq 2$ is $\mathbb{P}\left(c_a = x | (c_b)_{b<a}\right) \propto \min_{b<a} \|x - c_b\|^2$. This procedure thus also computes the optimal partition as a byproduct, like MAXMIN. It can be further extended in a greedy manner: at each step $a \geq 2$, $s$ candidates are sampled according to the previous probability distribution, the new SSE (with $a$ clusters) is computed, and the candidate with the lowest SSE is chosen. The number of candidates per step $s$ is usually logarithmic in the number of clusters $k$; in all our tests, we have used $s = \lfloor 2 + \log k \rfloor$.[2] We refer to the original variant as KM++ and to the greedy one as GKM++.

The computational complexity of KM++ is $O(kND)$ and it requires 1 NDC, like MAXMIN, while GKM++ scales like $O(kND \log k)$ and the NDC required are slightly less than the number of candidates $s$. In Celebi et al. (2013) GKM++ was reported as superior to KM++ and overall as one among the best linear (in $N$) stochastic methods; in Fränti & Sieranoja (2019) the results of KM++ were considered comparable to or slightly worse than MAXMIN; however, GKM++ was not tested.

**REF(INIT).** Bradley and Fayyad's "refine" seeding algorithm (Bradley & Fayyad, 1998) is the one that most resembles our proposed scheme. Indeed, the initial step of the two methods is basically the same, i.e. it consists in splitting the dataset into $J$ random subsets and clustering them individually with $k$-means, thus obtaining $J$ groups of $k$ centroids. The crucial difference relies in the way in which these $J$ solutions are merged, which in Bradley & Fayyad (1998) is referred to as a "smoothing" procedure: in the refine method, the whole pool of $Jk$ centroids is used as a new dataset and clustered for $J$ times. Each time, one of the previous centroid configurations is used as seed for the Lloyd algorithm. Out of the resulting $J$ configurations, the one with the smallest cost (computed on the pooled dataset) is finally chosen. Originally, the authors used UNIF as the seeding method for clustering the subsets, but this can be trivially generalized to other methods. We thus consider it a meta-method like PNN-smoothing, and denote it with REF(INIT) where INIT is the seeding method for the initial step.

If the computational cost of INIT is linear, $O(kND)$, then REF(INIT) scales as $O\left(\left(kN + Jk^2\right)D\right)$. The algorithm requires $J \leq N/k$ in order to perform the first step, and thus REF(INIT) is always at most $O(kND)$. In terms of NDC, the initial $J$ clusterings overall require at least the same amount as INIT (e.g. 1 if INIT is UNIF or MAXMIN) plus some additional ones for the Lloyd's iteration which are hard to estimate; then at least another $Jk/N$ NDC are needed for the merging; finally, 1 NDC is required at the end; overall, the lower bound on the NDC is $1 + Jk/N$ more than INIT. In the original publication, Bradley & Fayyad (1998), only REF(UNIF) with $J = 10$ was tested. This was also the setting used in Celebi et al. (2013); Fränti & Sieranoja (2019). We also used the same value of $J$ in our tests (the only exception being a dataset for which $N/k = 10$, in which case we used $J = 5$), but we tested more INIT algorithms. As mentioned above,

---

[2]This is the default value used by the scikit-learn library (Pedregosa et al., 2011) and seems to work well; Celebi et al. (2013) used $s = \log(k)$ (it's unclear if it was truncated or rounded).

in Celebi et al. (2013) REF(UNIF) was found to be among the best methods, whereas in Fränti & Sieranoja (2019) it was shown to perform rather poorly on synthetic datasets.

**PNN.** Our method starts out identically to REF(INIT), but it uses the PNN procedure to merge the resulting $J$ clusterings. This procedure was originally introduced in Equitz (1989) as a seeding algorithm. The original algorithm, which we call PNN, is hierarchical. It starts with $N$ clusters, one cluster per data point, and then merges pairs of clusters iteratively until only $k$ clusters remain. Merging two clusters means that the partition $\mathcal{P}$ is updated by substituting the two clusters with their union. The centroids set $\mathcal{C}$ is also updated, by substituting the two starting centroids $c_a$ and $c_b$ with the centroid of the new cluster, $c_{\text{new}} = (z_a c_a + z_b c_b) / (z_a + z_b)$, where $z_a$ and $z_b$ are the number of elements in each of the two original clusters. The algorithm is deterministic and greedy: the two clusters to be merged at each step are those whose merging will result in the smallest increase in the SSE cost. It is easy to see from eq. 1 that the cost increment of merging two clusters of sizes $z_a$ and $z_b$ and with centroids $c_a$ and $c_b$ is $\Delta_{ab} = z_a z_b \|c_a - c_b\|^2 / (z_a + z_b)$. Thus both the pairwise merging costs and the new centroid can be computed using only the centroids and the cluster sizes.

The initial computation of all the merging costs $\Delta_{ab}$ requires $N(N-1)/2$ distance computations, thus $O(DN^2)$ operations. The computational complexity of a merging step, assuming that we are going from $\hat{k}+1$ clusters to $\hat{k}$ clusters, would be $O(D\hat{k}^2)$ if performed straightforwardly, due to the need to update the $\Delta_{ab}$ after each merge. However, in Franti & Kaukoranta (1998) it was shown that a significant speedup can be obtained by considering that most cluster pairs are unaffected by individual merges, and the complexity can be reduced to $O(D\hat{k}\tau_{\hat{k}})$ where $\tau_{\hat{k}} \in \left[1, \hat{k}\right]$ is essentially the number of (potentially) affected clusters and is generally much smaller than $\hat{k}$. The number of distance computations of a merging step can easily be limited to exactly $\hat{k}$ by memoizing and keeping up-to-date all the distances; however, since $\tau_{\hat{k}}$ is usually very small, in practice this technique only helps the running time to a limited extent, and only in very high-dimensional cases, and thus for simplicity our implementation does not use it.[3] The update operations can be straightforwardly parallelized over $\hat{k}$, which proves advantageous above a certain threshold. In the original PNN algorithm the merging step must be performed $N - k$ times with $\hat{k}$ ranging from $N-1$ to $k$, which amounts at roughly $N^2/2$ distance computations (assuming $N \gg k$). Thus, the overall computational complexity is $\Omega(DN^2)$ and the NDC required are about $N/k$. This is quite expensive for large datasets. On the other hand, the results in terms of the SSE objective are generally very good.

The PNN scheme was also employed in the genetic algorithm of Fränti (2000), where however it was used as a crossover step rather than a seeding procedure. In that algorithm, two given configurations with $k$ clusters each are first merged into a single configuration with $2k$ clusters, which is then used as the starting point for the PNN iterative merging, until $k$ clusters remain. The resulting cost is $O(Dk^2\tau_k)$, which (crucially) does not involve a factor of $N$ thanks to the fact that only centroid computations are involved in the merge. Our seeding scheme is similar to this, in that we also use the iterative PNN merging procedure with an initial number of clusters much smaller than $N$.

## 3 The PNN-smoothing scheme

In this section we describe in detail the PNNS(INIT) seeding scheme and discuss its properties.

The inputs of the procedure are the same as for any other algorithm (the dataset $\mathcal{X}$ and the number of clusters $k$), plus the subset-seeding algorithm INIT, and one extra parameter $\rho$, used to determine the

---

[3]Significantly more distance computations could be skipped by keeping lower and upper bounds based on the triangle inequality, similarly to the approach used to accelerate Lloyd's iterations in Elkan (2003) and others. This appears to be a promising optimization for high-dimensional data, which is left for future work.

Table 1: Summary of seeding methods characteristics: computational complexity (only $N$ and $k$ dependence, all methods are linear in $D$) and normalized distance computations. For REF(INIT) and PNNS(INIT) only lower bounds on NDCs are available since the expressions don't account for the internal Lloyd's iterations. The $\tau_X$ terms for PNNS(INIT) and PNN represent hard-to-estimate terms, upper-bounded by $X$ but much smaller in practice. For GKM++ we set $s = \lfloor 2 + \log k \rfloor$. For REF(INIT) we normally set $J = 10$. For PNNS(INIT) we used the value $\rho = 1$ throughout the main text.

| method | complexity | NDCs |
|---|---|---|
| UNIF | $kN$ | 1 |
| MAXMIN | $kN$ | 1 |
| KM++ | $kN$ | 1 |
| GKM++ | $kNs$ | $1/k + s\,(k-1)\,/k$ |
| REF(INIT) | $kN$ | $\geq \mathrm{NDC}\,(\mathrm{INIT}) + 1 + Jk/N$ |
| PNNS(INIT) | $kN\tau_{\sqrt{kN}}$ | $\geq \mathrm{NDC}\,(\mathrm{INIT}) + 1 + \rho/2$ |
| PNN | $N^2\tau_N$ | $N/k$ |

number of subsets $J$. The output is a configuration to be used as a starting point for local optimization. The high-level summary (see also the illustration of each step in fig. 1) is as follows:

---

**Algorithm 1:** PNNS seeding

---

**Input:** $\mathcal{X}$, $k$, INIT, $\rho$

1. Set $J = \left\lceil \sqrt{\rho N/\,(2k)} \right\rceil$. Cap the result at $\lfloor N/k \rfloor$.

2. Split $\mathcal{X}$ into $J$ random subsets $\left(\tilde{\mathcal{X}}_a\right)_{a=1..J}$.

3. Cluster each $\tilde{\mathcal{X}}_a$ independently, using INIT for seeding followed by Lloyd's algorithm; obtain $J$ configurations $\left(\tilde{\mathcal{C}}_a, \tilde{\mathcal{P}}_a\right)$, with $k$ clusters each.

4. Collect the $J$ configurations $\left(\tilde{\mathcal{C}}_a, \tilde{\mathcal{P}}_a\right)$ into a single configuration $(\mathcal{C}_0, \mathcal{P}_0)$ for the entire $\mathcal{X}$, with $kJ$ centroids and clusters.

5. Merge the clusters of $(\mathcal{C}_0, \mathcal{P}_0)$, two at a time, using the PNN procedure, until $k$ clusters remain; obtain a set of $k$ centroids $\mathcal{C}$.

6. Compute the optimal partition $\mathcal{P}$ associated to $\mathcal{C}$.

---

**Output:** $(\mathcal{C}, \mathcal{P})$

---

Next, we describe and discuss in more detail each step.

1. The parameterization in terms of $\rho$ rather than $J$ ensures that the asymptotic behavior of the algorithm is almost linear in $N$ and $k$, as discussed in the introduction (cf. point 5 below). The upper bound to $J$ is necessary for the following steps 2 and 3. The effective range of $\rho$, assuming for simplicity that $N$ is divisible by $k$, is $[2k/N, 2N/k]$. The lower bound leads to $J = 1$ and PNNS(INIT)=INIT. The upper bound corresponds to $J = N/k$ and PNNS(INIT)=PNN (under the assumption that INIT invoked on $k$ points will assign each one to its own cluster). In other words, by changing $\rho$ we can interpolate between any seeding algorithm INIT and the pairwise-nearest-neighbor algorithm. This explains why, as a rule of thumb, increasing $\rho$ improves quality at the cost of performance, although this is not strictly true in all cases.

   The allowed range for $\rho$ is quite wide under normal circumstances, in which $k$ is much smaller than $N$. Throughout the main text we use the value $\rho = 1$, which from our preliminary analysis seems to result in a good trade-off in all cases and for all INIT algorithms. This is a valid (i.e. non-degenerate)

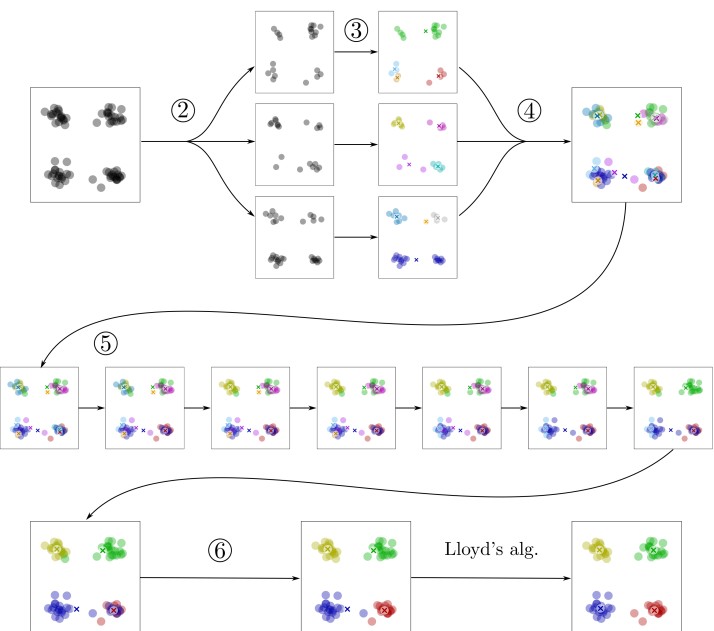

Figure 1: Example of PNNS(INIT) in action. Here $D = 2$, $N = 72$, $k = 4$ and INIT=UNIF. The numbering of the steps follows algorithm 1 (step 1 simply yields $J = 3$). The crosses represent centroids. After step 3 the subsets clusterings contain some clear mistakes, but after the merge (4) and the PNN procedure (5) the remaining 4 centroids are close to the correct positions. After step 6 the seeding procedure is completed; the final local optimization solves the problem.

choice whenever $k < N/2$, which is arguably always the case in realistic scenarios, thus making it a reasonable default value. Additional results with $\rho = 10$ are reported in Appendix A.

2. We split the data as evenly as possible, i.e. we create $N \mod J$ subsets of size $\lfloor N/J \rfloor + 1$ and $J - N \mod J$ subsets of size $\lfloor N/J \rfloor$. This is easy to implement efficiently by just constructing a sorted list of indices, each index being repeated for the appropriate number of times, and shuffling it. Our preliminary testing showed that the algorithm is not sensitive to the details of the splitting procedure.

3. The computational cost of each individual clustering of one of the subsets depends on the choice of INIT. Assuming that to be linear, like in the examples mentioned in the previous section, this scales like $N/J$ rather than $N$. Just like for refine, the NDC required for this step are at least the same as for INIT (thus at least 1) plus those for Lloyd's iterations, which are hard to estimate. This step is also trivially parallelizable.

4. In order to obtain the new configuration, we just take the union of the centroids, i.e. $\mathcal{C}_0 = \bigcup_a \tilde{\mathcal{C}}_a$; the partition $\mathcal{P}_0$ would also be simply the union of the $\left(\tilde{\mathcal{P}}_a\right)_a$ with remapped indices, but since it is not even needed for the algorithm (only the cluster sizes are used) it can be skipped. It is also interesting to note that the new configuration will, in general, be nowhere near optimal for a problem with $kJ$ clusters, since points near each other will likely be assigned to clusters coming from different subsets. Nevertheless, at least under some favorable scenarios, we can expect that the centroids in $\mathcal{C}_0$ may themselves be approximately clustered into $k$ groups (see fig. 1). This is the same intuition at the root of the refine method. The question then becomes how to best find a consensus configuration among the $J$ different results.

5. The PNN procedure will start, as the name implies, by merging the closest centroids (accounting for their associated cluster size). If the centroids in $\mathcal{C}_0$ are mostly clustered already, the procedure will

likely pick first the centroids that appeared in multiple subset clusterings. Each time two centroids are merged, their associated size (and thus weight) increases, such that even if in the last stages some very sub-optimal partitions are merged with a large cluster, the centroid will be heavily skewed toward the latter.

The computational cost of the merging scales like $O\left(D\left(Jk\right)^2 \tau_{Jk}\right) = O\left(DNk\tau_{Jk}\right)$, as per the analysis of the PNN procedure of the previous section. This is indeed a consequence of our choice for the scaling of $J$. The additional factor $\tau_{Jk}$ is hard to estimate; it is bounded by $Jk = \sqrt{\rho Nk/2}$ but in practice it appears to be quite small. This is the step that dominates the computational complexity of the whole algorithm; however, it is seldom the step that takes up the majority of computing time in practice when using $\rho = 1$. Indeed, the number of NDC can be estimated as just $\rho/2$ (where $\rho/4$ come from the initial step and $\rho/4$ from the merging process, analogously to the analysis presented for PNN).

6. At the end of the procedure the partition is recomputed (1 NDC). This is done to compute the SSE and provide a consistent starting point for Lloyd's algorithm (as discussed in the introduction), but it also means that outlier points in the partitions that could be (virtually) produced during the PNN merge are eliminated (like the blue points in the bottom-right cluster and the green points in the top-left cluster in fig. 1).

Overall, the number of NDC in the PNNS(INIT) procedure is larger by at least $1 + \rho/2$ compared to INIT (roughly, see steps 3, 5 and 6). This estimate appears to capture the biggest time penalty of PNNS(INIT) in practice, even though it does not account for the additional Lloyd's iterations performed in step 3. This is because those usually require much less than 1 NDC per iteration thanks to the use of accelerator algorithms; furthermore they are compensated by obtaining a seed which is closer to a local optimum, so that the overall number of Lloyd's iterations at the end of the process is comparable (the data showing this is reported in Appendix A).

## 4 Numerical experiments

### 4.1 Experimental setup

We performed a series of tests comparing PNNS with all the algorithms mentioned in sec. 2, namely: UNIF, MAXMIN, KM++, GKM++, PNN, and REF(INIT) and PNNS(INIT) with INIT∈ {UNIF, MAXMIN, KM++, GKM++}. We used the same data structures and programming language (Julia v1.7.3) for all of them. For the local optimization part, i.e. Lloyd's algorithm, we implemented a number of techniques that can accelerate the computation, while keeping it exact, by skipping some updates at the cost of some bookkeeping: the "reduced computation" method (RC) by Kaukoranta et al. (1999); the Elkan method (ELK) from Elkan (2003); the Hamerly method (HAM) from Hamerly (2010); the Yinyang method (YY) from Ding et al. (2015); the exponion method (EXP) from Newling & Fleuret (2016); the ball-kmeans method (BALL) from Xia et al. (2022). For ELK and YY we implemented the simplified versions described in Newling & Fleuret (2016). For each of the datasets that we tested we chose the accelerator technique that resulted in the fastest average convergence time, when using KM++ for seeding, across at least 30 random repetitions, and used that accelerator for all other tests with that dataset. The winner for each dataset is reported in Table 2. This choice puts PNNS at the maximum disadvantage, since, as a general tendency, the better the accelerator method, the larger the relative penalty of applying the PNNS scheme.[4] In all our tests Lloyd's algorithm was run until convergence to a fixed point. Our code is available at https://github.com/carlobaldassi/KMeansPNNSmoothing.jl.

All the timings that we report refer to tests performed on the same hardware (Intel Core i7-9750H 2.60GHz CPU with 6 physical cores, 64Gb DDR4 2666MHz RAM, running Ubuntu Linux 20.04 with 5.15.0 kernel)

---

[4]This is because, as mentioned above, the number of overall Lloyd's iteration ends up being comparable between INIT and PNNS(INIT), but accelerators become more effective as the iterations progress, and in PNNS Lloyd's scheme is effectively started twice (this effect is indeed captured by the NDC count discussed in sec. 3, point 6).

Table 2: Characteristics of the datasets used in the tests and best accelerator for each dataset.

| | dataset | $D$ | $N$ | $k$ | accel. |
|---|---|---|---|---|---|
| synthetic | *A3* | 2 | 7500 | 50 | EXP |
| | *Birch1* | 2 | 100000 | 100 | EXP |
| | *Birch2* | 2 | 100000 | 100 | EXP |
| | *Unbalance* | 2 | 6500 | 8 | EXP |
| | *Dim1024* | 1024 | 1024 | 16 | EXP |
| real-world | *Bridge* | 16 | 4096 | 256 | RC |
| | *House* | 3 | 34112 | 256 | EXP |
| | *Miss America* | 16 | 6480 | 256 | RC |
| | *UrbanGB* | 2 | 360177 | 469 | EXP |
| | *Olivetti* | 4096 | 400 | 40 | ELK |
| | *Isolet* | 617 | 7792 | 26 | ELK |
| | *USCensus* | 68 | 2458285 | 100 | YY |

with no other computationally intensive processes running while testing. All codes were carefully optimized[5] and can run in parallel with multi-threading (the parallelization is over the $N$ data points during Lloyd's iterations, and over the $\hat{k}$ centroids when $\hat{k} \geq 500$ during the PNN procedure, see sec. (2)); most of our results are shown for the single-threaded case except where otherwise noted. We also report the average total number of normalized distance computations (NDC), summing up those performed during seeding (discussed in the previous sections) and in the final optimization phase.

We tested a number of synthetic and real-world datasets whose characteristics are shown in table 2. The synthetic datasets are mainly intended to measure the ability of the algorithms to find the solution when one can be clearly identified, and for direct comparison with the results of ref. (Fränti & Sieranoja, 2019). In these kind of datasets, when *k*-means gets stuck in a sub-optimal minimum, it is usually due to having made one or more clearly identifiable mistakes (see e.g. the supplementary materials of ref. (Baldassi, 2022)). As we shall show in sec. 4.2, PNNS(INIT) is the only family of algorithms (among the linear or quasi-linear ones) that is capable of finding the solution in 100% of these cases; this gives a degree of confidence that, when using PNNS, *k*-means will not get stuck into clear and easily-fixed mistakes, and that any variation in the optimization result is more likely to emerge from the features of the datasets not being aligned with the SSE objective.

The real-world datasets on the other hand do not generally have a simple structure or a simple "solution", and their optimum SSE is unknown. Our tests in sec. 4.3 will show that the PNNS scheme can provide systematic improvements in the SSE with a bounded increase in computing time, generally offering a better trade-off than alternative methods.

## 4.2 Synthetic datasets

In Fränti & Sieranoja (2019), several synthetic datasets with different characteristics were chosen and tested in order to probe the strengths and weaknesses of several seeding algorithms with respect to properties of the data. We picked 5 of the most challenging ones, all obtained from the UEF repository (Fränti & Sieranoja, 2018): *A3*, *Birch1*, *Birch2*, *Unbalance* and *Dim1024* (see table 2). We scaled each dataset uniformly in order to make them span the range $[0, 1]^D$. The difficulty for *A3*, *Birch1* and *Birch2* is in their relatively large size and abundance of local minima; for *Unbalance*, it's the fact that some clusters are small and far from the bigger ones; for *Dim1024* it's the large dimensionality. All algorithms tested in Fränti & Sieranoja (2019) showed poor results in at least some of these datasets; in particular, the authors report a 0% success rate (as defined below) on *Birch1* for all algorithms.

---

[5]Our implementations of ELK, HAM, YY and EXP are generally roughly comparable to the very optimized ones (written in C++) that accompany Newling & Fleuret (2016), available at https://github.com/idiap/eakmeans; our implementation of BALL is generally faster than the C++ one provided by the original authors of Xia et al. (2022), available at https://github.com/syxiaa/ball-k-means.

Table 3: Results on synthetic datasets

success rate

|  | A3 | Birch1 | Birch2 | Unbalance | Dim1024 |
|---|---|---|---|---|---|
| UNIF | 0 | 0 | 0 | 0 | 0.001 |
| MAXMIN | 0.004 | 0 | 0 | 0.223 | 1 |
| KM++ | 0 | 0 | 0 | 0.541 | 0.996 |
| GKM++ | 0.051 | 0.02 | 0.06 | 0.946 | 1 |
| REF(UNIF) | 0 | 0 | 0 | 0 | 0.055 |
| REF(GKM++) | 0.239 | 0.03 | 0.44 | 1 | 1 |
| PNNS(UNIF) | 0.709 | 1 | 0.26 | 0.696 | 0.919 |
| PNNS(MAXMIN) | 1 | 1 | 1 | 1 | 1 |
| PNNS(KM++) | 0.98 | 1 | 1 | 1 | 1 |
| PNNS(GKM++) | 1 | 1 | 1 | 1 | 1 |
| PNN | 1 | 1 | 1 | 1 | 1 |

convergence time (mean±stdev)

|  | A3 (in $10^{-3}s$) | Birch1 (in $10^{-1}s$) | Birch2 (in $10^{-1}s$) | Unbalance (in $10^{-3}s$) | Dim1024 (in $10^{-3}s$) |
|---|---|---|---|---|---|
| UNIF | $5.1 \pm 1.5$ | $2.3 \pm 0.6$ | $0.63 \pm 0.10$ | $3.1 \pm 1.8$ | $5.5 \pm 1.3$ |
| MAXMIN | $4.0 \pm 0.9$ | $2.0 \pm 0.4$ | $0.75 \pm 0.10$ | $0.7 \pm 0.8$ | $5.4 \pm 0.8$ |
| KM++ | $4.1 \pm 1.2$ | $1.78 \pm 0.38$ | $0.66 \pm 0.10$ | $1.1 \pm 1.3$ | $5.4 \pm 1.5$ |
| GKM++ | $6.5 \pm 1.4$ | $2.44 \pm 0.26$ | $1.64 \pm 0.11$ | $0.9 \pm 1.0$ | $12.8 \pm 1.7$ |
| REF(UNIF) | $14.7 \pm 2.5$ | $2.97 \pm 0.33$ | $1.39 \pm 0.09$ | $4.7 \pm 1.5$ | $20.5 \pm 3.4$ |
| REF(GKM++) | $13.8 \pm 2.2$ | $3.06 \pm 0.23$ | $2.06 \pm 0.12$ | $2.0 \pm 1.3$ | $19.1 \pm 3.0$ |
| PNNS(UNIF) | $11.2 \pm 2.0$ | $2.79 \pm 0.09$ | $1.78 \pm 0.11$ | $3.7 \pm 1.6$ | $14.2 \pm 2.9$ |
| PNNS(MAXMIN) | $8.4 \pm 1.7$ | $2.51 \pm 0.10$ | $1.60 \pm 0.09$ | $1.8 \pm 1.4$ | $12.9 \pm 2.5$ |
| PNNS(KM++) | $9.1 \pm 1.8$ | $2.55 \pm 0.07$ | $1.59 \pm 0.07$ | $2.0 \pm 1.2$ | $13.1 \pm 2.8$ |
| PNNS(GKM++) | $12.1 \pm 2.0$ | $3.32 \pm 0.08$ | $2.49 \pm 0.09$ | $2.5 \pm 1.4$ | $19.9 \pm 3.3$ |
| PNN | $396 \pm 16$ | $660 \pm 6$ | $709.6 \pm 3.1$ | $292 \pm 11$ | $1678 \pm 34$ |

normalized distance computations (mean±stdev)

|  | A3 | Birch1 | Birch2 | Unbalance | Dim1024 |
|---|---|---|---|---|---|
| UNIF | $1.80 \pm 0.18$ | $2.52 \pm 0.37$ | $1.146 \pm 0.027$ | $2.7 \pm 0.6$ | $1.89 \pm 0.32$ |
| MAXMIN | $1.33 \pm 0.09$ | $2.08 \pm 0.29$ | $1.073 \pm 0.020$ | $1.26 \pm 0.10$ | $1.07812 \pm 0.0$ |
| KM++ | $1.41 \pm 0.12$ | $2.00 \pm 0.25$ | $1.063 \pm 0.014$ | $1.35 \pm 0.31$ | $1.079 \pm 0.009$ |
| GKM++ | $5.08 \pm 0.05$ | $6.46 \pm 0.16$ | $5.972 \pm 0.004$ | $3.78 \pm 0.11$ | $3.89062 \pm 0.0$ |
| REF(UNIF) | $3.94 \pm 0.12$ | $3.75 \pm 0.21$ | $2.393 \pm 0.019$ | $4.6 \pm 0.5$ | $6.6 \pm 0.4$ |
| REF(GKM++) | $6.992 \pm 0.029$ | $7.72 \pm 0.11$ | $7.1128 \pm 0.0030$ | $5.068 \pm 0.028$ | $6.9261 \pm 0.0014$ |
| PNNS(UNIF) | $3.84 \pm 0.06$ | $4.07 \pm 0.04$ | $3.191 \pm 0.012$ | $4.63 \pm 0.37$ | $4.27 \pm 0.16$ |
| PNNS(MAXMIN) | $3.283 \pm 0.025$ | $3.662 \pm 0.025$ | $3.044 \pm 0.007$ | $3.257 \pm 0.026$ | $3.380 \pm 0.027$ |
| PNNS(KM++) | $3.439 \pm 0.039$ | $3.791 \pm 0.031$ | $3.084 \pm 0.008$ | $3.40 \pm 0.05$ | $3.380 \pm 0.027$ |
| PNNS(GKM++) | $7.137 \pm 0.022$ | $8.416 \pm 0.020$ | $7.957 \pm 0.007$ | $5.903 \pm 0.034$ | $6.193 \pm 0.027$ |
| PNN | $262.324$ | $1730.39$ | $1738.25$ | $1422.21$ | $531.759$ |

Table 4: Results on real-world datasets

mean SSE cost (mean±stdev over 100 repetitions)

| | Bridge ($\times 10^7$) | House ($\times 10^5$) | Miss A. ($\times 10^5$) | Urb.GB ($\times 10^2$) | Olivetti ($\times 10^4$) | Isolet ($\times 10^5$) |
|---|---|---|---|---|---|---|
| UNIF | $1.178 \pm 0.009$ | $10.11 \pm 0.13$ | $6.07 \pm 0.05$ | $6.8 \pm 1.1$ | $1.296 \pm 0.027$ | $1.196 \pm 0.010$ |
| MAXMIN | $1.139 \pm 0.005$ | $10.18 \pm 0.07$ | $5.794 \pm 0.028$ | $2.99 \pm 0.07$ | $1.252 \pm 0.015$ | $1.233 \pm 0.014$ |
| KM++ | $1.154 \pm 0.007$ | $9.60 \pm 0.04$ | $5.687 \pm 0.037$ | $2.72 \pm 0.06$ | $1.276 \pm 0.021$ | $1.196 \pm 0.010$ |
| GKM++ | $1.124 \pm 0.004$ | $9.529 \pm 0.027$ | $5.507 \pm 0.016$ | $2.430 \pm 0.020$ | $1.227 \pm 0.013$ | $1.190 \pm 0.007$ |
| REF(UNIF) | $1.162 \pm 0.006$ | $9.95 \pm 0.10$ | $5.850 \pm 0.035$ | $5.3 \pm 0.4$ | $1.298 \pm 0.023$ | $1.189 \pm 0.006$ |
| REF(GKM++) | $1.159 \pm 0.005$ | $9.562 \pm 0.027$ | $5.826 \pm 0.031$ | $2.420 \pm 0.022$ | $1.255 \pm 0.020$ | $1.1837 \pm 0.0036$ |
| PNNS(UNIF) | $1.124 \pm 0.005$ | $9.553 \pm 0.034$ | $5.607 \pm 0.040$ | $2.70 \pm 0.06$ | $1.215 \pm 0.014$ | $1.1800 \pm 0.0019$ |
| PNNS(KM++) | $1.1076 \pm 0.0033$ | $9.486 \pm 0.020$ | $5.403 \pm 0.013$ | $2.323 \pm 0.010$ | $1.209 \pm 0.013$ | $1.1795 \pm 0.0016$ |
| PNNS(GKM++) | $1.0947 \pm 0.0029$ | $9.476 \pm 0.020$ | $5.342 \pm 0.010$ | $2.297 \pm 0.004$ | $1.189 \pm 0.007$ | $1.1790 \pm 0.0015$ |
| PNN | $1.08279$ | $9.49701$ | $5.31588$ | $2.3153$ | $1.16238$ | $1.17692$ |

minimum SSE cost (over 100 repetitions)

| | Bridge ($\times 10^7$) | House ($\times 10^5$) | Miss A. ($\times 10^5$) | Urb.GB ($\times 10^2$) | Olivetti ($\times 10^4$) | Isolet ($\times 10^5$) |
|---|---|---|---|---|---|---|
| UNIF | $1.15723$ | $9.87713$ | $5.95536$ | $5.42031$ | $1.23747$ | $1.1795$ |
| MAXMIN | $1.12869$ | $10.0159$ | $5.72134$ | $2.90322$ | $1.21735$ | $1.20639$ |
| KM++ | $1.14219$ | $9.52638$ | $5.59949$ | $2.60788$ | $1.23035$ | $1.18025$ |
| GKM++ | $1.11421$ | $9.45992$ | $5.47583$ | $2.37869$ | $1.19908$ | $1.17709$ |
| REF(UNIF) | $1.14562$ | $9.77128$ | $5.76604$ | $4.39285$ | $1.24007$ | $1.17688$ |
| REF(GKM++) | $1.14344$ | $9.49776$ | $5.74112$ | $2.37821$ | $1.2116$ | $1.17737$ |
| PNNS(UNIF) | $1.11345$ | $9.47461$ | $5.51748$ | $2.5921$ | $1.17155$ | $1.17685$ |
| PNNS(KM++) | $1.10041$ | $9.44324$ | $5.37209$ | $2.30731$ | $1.18579$ | $1.17687$ |
| PNNS(GKM++) | $1.08729$ | $9.42885$ | $5.31768$ | $2.28822$ | $1.17066$ | $1.17695$ |
| PNN | $1.08279$ | $9.49701$ | $5.31588$ | $2.3153$ | $1.16238$ | $1.17692$ |

convergence time (mean±stdev)

| | Bridge (in $10^{-2}s$) | House (in $10^{-1}s$) | Miss A. (in $10^{-1}s$) | Urb.GB (in $s$) | Olivetti (in $10^{-2}s$) | Isolet (in $10^{-1}s$) |
|---|---|---|---|---|---|---|
| UNIF | $4.17 \pm 0.38$ | $3.3 \pm 0.6$ | $1.13 \pm 0.09$ | $1.63 \pm 0.23$ | $2.7 \pm 0.6$ | $1.52 \pm 0.30$ |
| MAXMIN | $4.6 \pm 0.5$ | $3.2 \pm 0.5$ | $1.78 \pm 0.33$ | $1.73 \pm 0.20$ | $4.1 \pm 0.6$ | $1.58 \pm 0.21$ |
| KM++ | $4.22 \pm 0.36$ | $2.40 \pm 0.37$ | $1.15 \pm 0.11$ | $1.56 \pm 0.13$ | $4.10 \pm 0.39$ | $1.75 \pm 0.29$ |
| GKM++ | $6.45 \pm 0.35$ | $3.33 \pm 0.33$ | $1.58 \pm 0.13$ | $5.51 \pm 0.16$ | $8.8 \pm 0.6$ | $3.02 \pm 0.29$ |
| REF(UNIF) | $22.0 \pm 0.5$ | $6.5 \pm 0.4$ | $3.26 \pm 0.11$ | $3.65 \pm 0.28$ | $8.0 \pm 0.5$ | $1.68 \pm 0.24$ |
| REF(GKM++) | $24.6 \pm 0.6$ | $6.48 \pm 0.36$ | $3.99 \pm 0.19$ | $5.36 \pm 0.23$ | $12.0 \pm 0.4$ | $2.19 \pm 0.22$ |
| PNNS(UNIF) | $5.90 \pm 0.31$ | $4.74 \pm 0.31$ | $1.52 \pm 0.09$ | $4.12 \pm 0.17$ | $6.2 \pm 1.0$ | $1.63 \pm 0.16$ |
| PNNS(KM++) | $5.9 \pm 0.4$ | $4.44 \pm 0.31$ | $1.52 \pm 0.10$ | $3.58 \pm 0.16$ | $6.74 \pm 0.38$ | $1.66 \pm 0.17$ |
| PNNS(GKM++) | $8.13 \pm 0.39$ | $5.22 \pm 0.32$ | $1.89 \pm 0.11$ | $5.99 \pm 0.14$ | $9.51 \pm 0.39$ | $2.07 \pm 0.18$ |
| PNN | $36.2 \pm 1.2$ | $93.68 \pm 0.20$ | $9.356 \pm 0.033$ | $819 \pm 7$ | $30.5 \pm 1.8$ | $297.2 \pm 2.8$ |

normalized distance computations (mean±stdev)

| | Bridge | House | Miss A. | Urb.GB | Olivetti | Isolet |
|---|---|---|---|---|---|---|
| UNIF | $8.4 \pm 0.5$ | $5.7 \pm 0.9$ | $13.5 \pm 0.9$ | $1.60 \pm 0.23$ | $1.79 \pm 0.14$ | $3.35 \pm 0.36$ |
| MAXMIN | $8.9 \pm 0.8$ | $4.5 \pm 0.6$ | $23 \pm 4$ | $1.196 \pm 0.037$ | $2.21 \pm 0.07$ | $3.14 \pm 0.25$ |
| KM++ | $8.2 \pm 0.6$ | $3.71 \pm 0.39$ | $13.7 \pm 1.3$ | $1.229 \pm 0.029$ | $2.26 \pm 0.07$ | $3.36 \pm 0.35$ |
| GKM++ | $13.2 \pm 0.5$ | $9.27 \pm 0.32$ | $19.5 \pm 1.5$ | $8.168 \pm 0.033$ | $6.04 \pm 0.05$ | $6.92 \pm 0.33$ |
| REF(UNIF) | $44.8 \pm 0.8$ | $8.3 \pm 0.6$ | $39.4 \pm 1.0$ | $3.15 \pm 0.15$ | $5.68 \pm 0.10$ | $4.62 \pm 0.26$ |
| REF(GKM++) | $49.5 \pm 0.8$ | $12.25 \pm 0.35$ | $48.3 \pm 2.1$ | $9.465 \pm 0.021$ | $10.34 \pm 0.07$ | $8.20 \pm 0.22$ |
| PNNS(UNIF) | $10.81 \pm 0.37$ | $6.80 \pm 0.36$ | $16.6 \pm 0.9$ | $4.03 \pm 0.09$ | $4.25 \pm 0.10$ | $5.06 \pm 0.15$ |
| PNNS(KM++) | $10.6 \pm 0.4$ | $6.05 \pm 0.29$ | $16.7 \pm 1.0$ | $3.31 \pm 0.06$ | $5.00 \pm 0.10$ | $5.24 \pm 0.17$ |
| PNNS(GKM++) | $15.9 \pm 0.4$ | $11.59 \pm 0.30$ | $22.2 \pm 1.2$ | $10.191 \pm 0.014$ | $8.76 \pm 0.09$ | $8.73 \pm 0.16$ |
| PNN | $51.8177$ | $271.383$ | $87.8464$ | $1351$ | $19.7431$ | $739.388$ |

All of these datasets were generated from isotropic Gaussians centered around ground-truth centroids, and for all of them the global optimum of the SSE is very close to the ground truth. Under these circumstances, it is reasonable to classify the local minima configurations that the algorithms produce by their "centroid index" (CI), as defined in Fränti et al. (2014). The CI is computed by matching each centroid of a clustering with its closest one from the ground truth, and counting the number of unmatched ground truth centroids; in formulas:

$$\text{CI} = \sum_{b=1}^{k^{\text{gt}}} \mathbb{1} \left( \nexists a \in \{1, \dots, k\} \,:\, b = \underset{1 \le b' \le k^{\text{gt}}}{\arg \min} \, d \left( c_a, c_{b'}^{\text{gt}} \right)^2 \right) \tag{2}$$

where $c_b^{\text{gt}}$ with $b \in \{1, \dots, k^{\text{gt}}\}$ are the ground-truth centroids (in our setup $k = k^{\text{gt}}$), and $\mathbb{1}(\cdot)$ is an indicator function. The CI can be interpreted as the number of mistakes in the resulting clustering, and therefore we define the success rate of an algorithm as the frequency with which it finds a solution, i.e. a configuration with $\text{CI} = 0$.

We present our most representative results in table 3; the complete results are reported in Appendix A.1. All algorithms except for PNN and those in the PNNS family fail badly in at least some dataset. Conversely, all the PNNS(INIT) algorithms solve all the datasets in 100% of the cases (98% for PNNS(KM++) on *A3*), with the only notable exception being INIT=UNIF. Even in that case, however, the performance is much better than UNIF and REF(UNIF), and at times even than REF(GKM++) (which is the variant in the refine family that gives the best overall results). This demonstrates that (at least for this scenario) PNN-smoothing is a considerably better smoothing technique than refine.

The timings of the PNNS(INIT) algorithms are generally comparable with, and often better than, those of the refine family, and within a constant factor (smaller than 3) of the basic methods; on the other hand, the only other algorithm capable of solving all the datasets, PNN, is $\Omega\left(N^2\right)$ and indeed orders of magnitude slower. These conclusions are corroborated by inspecting the NDC, which depend on the accelerator used but not on the implementation or the hardware, and are roughly correlated with the running time.

Overall, our results show that the PNNS(INIT) scheme is able to improve the success rate of any INIT algorithm at the cost of a modest time penalty, and that it is a considerably better at achieving this than the alternatives. It is particularly interesting to consider the case of the *Birch1* dataset, for which UNIF, MAXMIN and KM++ all have a 0% success rate. Both GKM++ and REF(KM++) are intended to improve KM++ at the cost of additional computations, and REF(GKM++) combines the two approaches; yet, none of them improves the success rate beyond 3%. The PNNS(KM++) method, on the other hand, achieves 100% success rate on *Birch1* in the same or smaller amount of time as those other methods (and fewer NDCs).

### 4.3 Real-world datasets

We tested 7 challenging real-world datasets (see table 2). The first three, *Bridge*, *House* and *Miss America*, were obtained from the UEF repository (Fränti & Sieranoja, 2018); *UrbanGB*, *Isolet* and *USCensus* are from the UCI repository (Dua & Graff, 2017) and *Olivetti* from scikit-learn (Pedregosa et al., 2011). All of these are comparatively large and quite challenging for SSE optimization. The *Isolet* dataset was chosen as the hardest one among those tested in Celebi et al. (2013) (based on the results reported there); the *USCensus* dataset was chosen for its large size, to provide a test of parallelization efficiency; the other 5 datasets were also tested in Baldassi (2022), where it was shown that even sophisticated, state-of-the-art evolutionary algorithms cannot easily find their global minima (and indeed it is not even clear if they can find them at all). We did not scale the datasets, except *UrbanGB* for which we scaled the longitude by a factor of 1.7 to make distances roughly proportional to geographical distances, and *USCensus* for which we linearly transformed each dimension individually to make them fit into the range $[-1, 1]$. For all datasets, except *UrbanGB* and *Isolet*, the choice of $k$ follows existing literature and is otherwise arbitrary.

We performed 100 tests for each dataset (30 for *UrbanGB* and *USCensus*) and computed the average and minimum SSE cost achieved across the runs (the latter metric can provide an indication about what can be achieved with a multiple-restarts strategy by each algorithm), as well as the average running time, NDCs, Lloyd's iterations. As for the previous section, we report our most representative results here; the full results

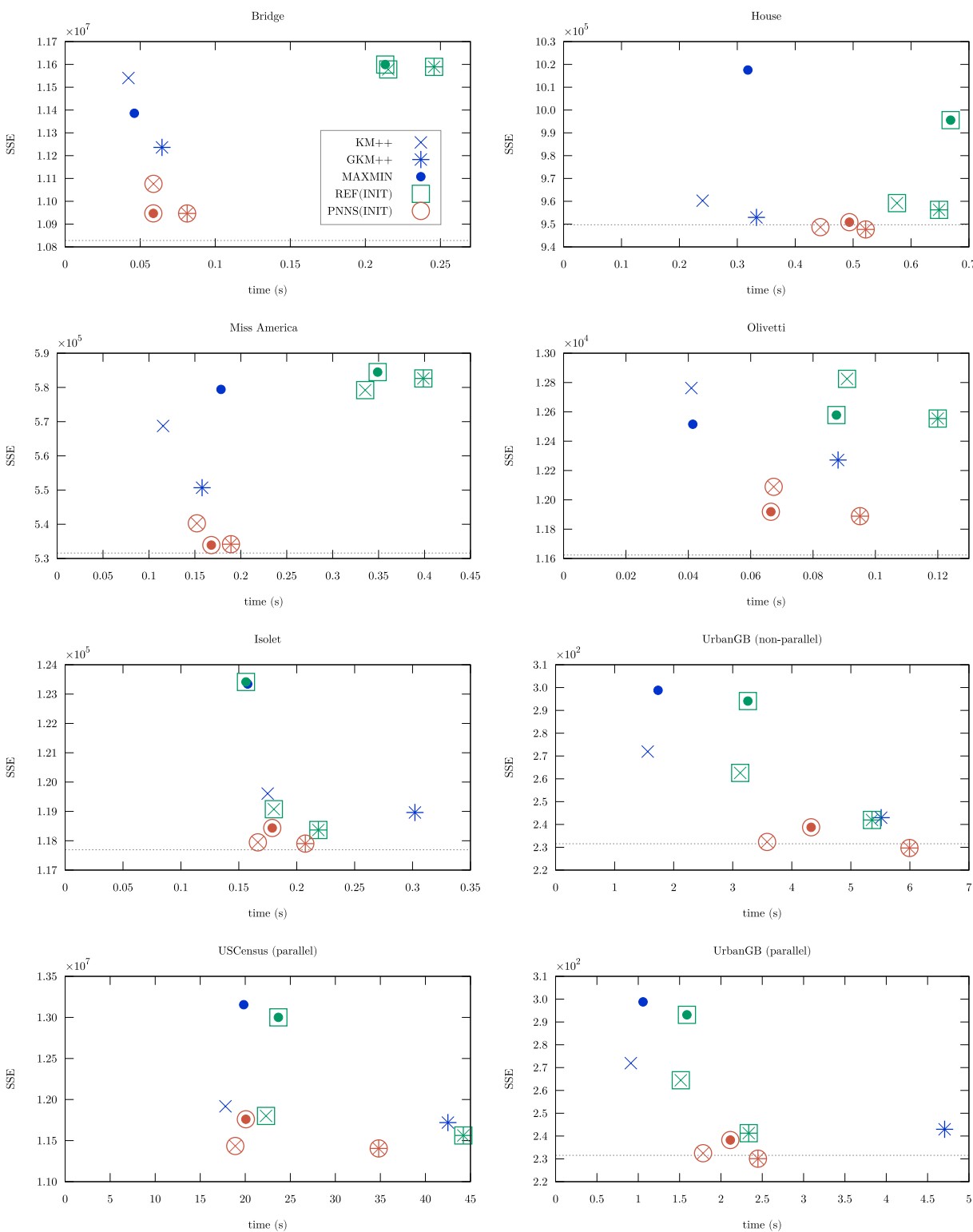

Figure 2: SSE cost vs convergence time, averages over 100 samples (only 30 for *UrbanGB* and *USCensus*), for the real-world datasets. The meta-methods symbols enclose the symbol for their INIT algorithm. The dashed lines are the costs achieved by PNN, whose timings would be off-scale. The top 6 panels are non-parallel test; the bottom row are tests performed with 4 threads in parallel.

can be found in Appendices A.2 and A.3. We first present the results for the non-parallel case (on all datasets except *USCensus*). Some results are presented in table 4, confirming that the PNNS family of algorithms attains results superior to all other linear algorithms, in comparable times (and comparable NDC), both in terms of the average and of the minimum SSE cost. More specifically, PNNS(INIT) consistently achieves better costs than both INIT and REF(INIT), for all datasets and all the tested INIT (note that REF(INIT) is not consistently better than INIT). This is confirmed by a statistical significance analysis (standard two-sided Wilcoxon rank-sum test, all $p$-values smaller than $10^{-11}$); in fact, in nearly all cases the average cost for PNNS(INIT) is even smaller than the minimum cost for INIT. In many cases, the average time for PNNS(INIT) is shorter than for REF(INIT). PNNS(GKM++) generally achieves the best SSEs (possibly on par with other PNNS methods), and in 2 cases out of 6 it even outperforms PNN, which belongs to a different computational class and is considerably slower.

In fig. 2, we plot the mean SSE vs convergence time for the best linear "plain" methods, KM++, GKM++ and MAXMIN, their REF(INIT) versions[6], and their PNNS(INIT) versions. They all have comparable timings, but the PNNS(INIT) family is consistently below the others. The dashed lines denote for reference the costs achieved by PNN (its timings would all be off-scale, cf. table 4), showing that even when it performs better than the algorithms in the PNNS family, the difference is generally relatively small.

In order to test the effect of parallelization on the performance of the algorithm, we also performed some tests with multi-threading enabled, using 4 threads. We tested the two largest datasets, *USCensus* and *UrbanGB*. The results are shown in the last two rows of fig. 2 (complete data in Appendix A.3); the case of *UrbanGB* allows a comparison with the non-parallel ones; note that we did not run PNN on *USCensus* because it would be impractical. The results are qualitatively similar to the non-parallel versions.

As for the synthetic datasets, it is particularly interesting to compare the different existing methods that can be used to improve on KM++, namely GKM++, REF(KM++) and REF(GKM++), with our method PNNS(KM++). Our method achieves better SSE costs than all of the others; in 5 out of 7 datasets it is also faster (for *House* only GKM++ is faster, for *UrbanGB* only REF(KM++) is faster). Even in this real-world setting, our PNNS scheme provides a better trade-off between cost and computational time than the existing alternatives.

## 5 Discussion

We have presented a scheme for $k$-means seeding called PNN-smoothing that can be applied to any existing linear ($O(kND)$) algorithm, with a limited impact on the scaling and on the convergence times. Our experiments, performed with an efficient implementation and state-of-the-art techniques, show clear and consistent improvements on challenging synthetic and real-world datasets, and systematically superior results (both in terms of quality and of speed) with respect to the similar "refine" smoothing scheme (also note that to the best of our knowledge we were the first to report tests for REF(INIT) with INIT$\neq$UNIF, finding that it does not systematically improve over INIT). One particularly interesting case is that of the very popular $k$-means++ seeding algorithm: we showed that our scheme outperforms alternative enhancement techniques (i.e. making it "greedy", using "refine", or both) in terms of both quality and speed.[7]

The overall picture is unchanged in a parallel multi-threading context. We also verified that, as one would expect, the results are qualitatively the same regardless of the scheme used to accelerate $k$-means iteration. The experiments also indicate that PNN-smoothing is not particularly susceptible to the original seeding scheme.

Overall, our results do not highlight any clear best among the PNNS algorithms: PNNS(KM++) is ususally the fastest, and it produces consistently good results, but PNNS(MAXMIN) is not much different, while the best costs are usually obtained by PNNS(GKM++). When trying to improve the cost sacrificing performance in difficult cases, the best strategy is probably to use PNNS(GKM++) with a larger $\rho$. From the results reported above, one can obtain an estimate of the potential gain of increasing $\rho$ by looking at the difference

---

[6]For the Olivetti dataset $N/k = 10$ and thus we used $J = 5$ instead of the default $J = 10$.

[7]At the time of writing, greedy-$k$-means++ is the default seeding method in the very popular scikit-learn package; according to our results, PNNS(KM++) would be a superior default.

between the costs achieved with $\rho = 1$ and those achieved with PNN. Since this difference is rather small,[8] one would not expect a large improvement (in Appendix A we report the results with $\rho = 10$). Another option would be to apply the PNN-smoothing scheme recursively (e.g. PNNS(PNNS(KM++)) and similar): we thoroughly explored this possibility,[9] but concluded that it does not bring significant advantages over simply using a larger $\rho$. On the other hand, if minimizing the costs is of paramount importance, there exist better optimization schemes than simple $k$-means, most notably evolutionary algorithms that use $k$-means as their starting point (Fränti, 2000; Baldassi, 2022); for those, the seeding process is comparatively less important in general, but it can still be crucial in some cases (as discussed in ref. (Baldassi, 2022), when trying to access low-cost regions of the configuration space, improving the seeding is usually more efficient than blind exploration via random mutations). Assessing the effect of better seeding on algorithmic schemes that go beyond simple $k$-means (like the above-mentioned evolutionary schemes, or for scenarios in which full Lloyd's iterations are too costly, or data cannot fit into memory, etc.) is left for future work.

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

# A  Appendix: Complete numerical results

In this section we report the full data for all numerical tests of sec. 4 of the main text. This complements tables 3 and 4 of the main text, and includes additional measures. When we report the average number of Lloyd's iterations, we include the ones preformed during seeding by REFINE(INIT) and PNNS(INIT), but we scale them by a factor of $J$ to keep them comparable to the ones performed during the final optimization.

Whenever we report means, we also report the standard deviations.

## A.1  Synthetic datasets

We tested 1000 samples for *A3*, *Unbalance* and *Dim1024*, and 100 samples for *Birch1* and *Birch2*.

### A.1.1  Convergence time (in s)

| seeder | $A3$ $(\times 10^{-3})$ | $Birch1$ $(\times 10^{-1})$ | $Birch2$ $(\times 10^{-1})$ | $Unbalance$ $(\times 10^{-3})$ | $Dim1024$ $(\times 10^{-3})$ |
|---|---|---|---|---|---|
| UNIF | $5.1 \pm 1.5$ | $2.3 \pm 0.6$ | $0.63 \pm 0.10$ | $3.1 \pm 1.8$ | $5.5 \pm 1.3$ |
| MAXMIN | $4.0 \pm 0.9$ | $2.0 \pm 0.4$ | $0.75 \pm 0.10$ | $0.7 \pm 0.8$ | $5.4 \pm 0.8$ |
| KM++ | $4.1 \pm 1.2$ | $1.78 \pm 0.38$ | $0.66 \pm 0.10$ | $1.1 \pm 1.3$ | $5.4 \pm 1.5$ |
| GKM++ | $6.5 \pm 1.4$ | $2.44 \pm 0.26$ | $1.64 \pm 0.11$ | $0.9 \pm 1.0$ | $12.8 \pm 1.7$ |
| REF(UNIF) | $14.7 \pm 2.5$ | $2.97 \pm 0.33$ | $1.39 \pm 0.09$ | $4.7 \pm 1.5$ | $20.5 \pm 3.4$ |
| REF(MAXMIN) | $10.2 \pm 2.5$ | $2.40 \pm 0.21$ | $1.25 \pm 0.11$ | $1.5 \pm 1.1$ | $13.8 \pm 2.8$ |
| REF(KM++) | $12.1 \pm 2.7$ | $2.49 \pm 0.25$ | $1.28 \pm 0.10$ | $1.8 \pm 1.2$ | $13.6 \pm 2.9$ |
| REF(GKM++) | $13.8 \pm 2.2$ | $3.06 \pm 0.23$ | $2.06 \pm 0.12$ | $2.0 \pm 1.3$ | $19.1 \pm 3.0$ |
| PNNS(UNIF) | $11.2 \pm 2.0$ | $2.79 \pm 0.09$ | $1.78 \pm 0.11$ | $3.7 \pm 1.6$ | $14.2 \pm 2.9$ |
| PNNS(MAXMIN) | $8.4 \pm 1.7$ | $2.51 \pm 0.10$ | $1.60 \pm 0.09$ | $1.8 \pm 1.4$ | $12.9 \pm 2.5$ |
| PNNS(KM++) | $9.1 \pm 1.8$ | $2.55 \pm 0.07$ | $1.59 \pm 0.07$ | $2.0 \pm 1.2$ | $13.1 \pm 2.8$ |
| PNNS(GKM++) | $12.1 \pm 2.0$ | $3.32 \pm 0.08$ | $2.49 \pm 0.09$ | $2.5 \pm 1.4$ | $19.9 \pm 3.3$ |
| PNNS(UNIF; $\rho = 10$) | $30.0 \pm 1.9$ | $6.60 \pm 0.16$ | $5.59 \pm 0.11$ | $5.2 \pm 2.0$ | $50.9 \pm 2.5$ |
| PNNS(MAXMIN; $\rho = 10$) | $26.1 \pm 3.5$ | $6.12 \pm 0.12$ | $5.19 \pm 0.12$ | $3.7 \pm 0.8$ | $54.9 \pm 2.0$ |
| PNNS(KM++; $\rho = 10$) | $26.9 \pm 1.0$ | $6.26 \pm 0.06$ | $5.31 \pm 0.07$ | $3.9 \pm 0.7$ | $55.3 \pm 2.1$ |
| PNNS(GKM++; $\rho = 10$) | $31.8 \pm 1.2$ | $7.07 \pm 0.21$ | $6.24 \pm 0.17$ | $4.8 \pm 0.8$ | $60.9 \pm 2.0$ |
| PNN | $396 \pm 16$ | $660 \pm 6$ | $709.6 \pm 3.1$ | $292 \pm 11$ | $1678 \pm 34$ |

### A.1.2  SSE cost

| seeder | $A3$ | $Birch1$ | $Birch2$ | $Unbalance$ | $Dim1024$ |
|---|---|---|---|---|---|
| UNIF | $11.5 \pm 1.4$ | $110 \pm 4$ | $1.56 \pm 0.22$ | $6.3 \pm 1.1$ | $(1.4 \pm 0.4) \cdot 10^4$ |
| MAXMIN | $8.5 \pm 0.6$ | $106.5 \pm 3.2$ | $0.84 \pm 0.10$ | $6 \pm 4$ | $5.3938 \pm 0.0$ |
| KM++ | $9.4 \pm 0.9$ | $104.7 \pm 3.0$ | $0.85 \pm 0.10$ | $1.1 \pm 0.6$ | $20 \pm 220$ |
| GKM++ | $7.7 \pm 0.5$ | $99.5 \pm 2.7$ | $0.528 \pm 0.035$ | $0.68 \pm 0.16$ | $5.3938 \pm 0.0$ |
| REF(UNIF) | $9.7 \pm 0.8$ | $104.0 \pm 2.8$ | $1.22 \pm 0.11$ | $4.5 \pm 1.3$ | $(5.3 \pm 2.4) \cdot 10^3$ |
| REF(MAXMIN) | $7.17 \pm 0.28$ | $100.4 \pm 1.9$ | $0.56 \pm 0.05$ | $0.646935 \pm 0.0$ | $5.3938 \pm 0.0$ |
| REF(KM++) | $8.4 \pm 0.5$ | $101.0 \pm 2.5$ | $0.71 \pm 0.05$ | $0.648 \pm 0.030$ | $5.3938 \pm 0.0$ |
| REF(GKM++) | $7.16 \pm 0.26$ | $96.7 \pm 1.7$ | $0.481 \pm 0.021$ | $0.646935 \pm 0.0$ | $5.3938 \pm 0.0$ |
| PNNS(UNIF) | $6.90 \pm 0.27$ | $92.77290 \pm 0.00006$ | $0.497 \pm 0.030$ | $1.0 \pm 0.6$ | $(3 \pm 9) \cdot 10^2$ |
| PNNS(MAXMIN) | $6.73784 \pm 0.00012$ | $92.77290 \pm 0.00005$ | $0.456724 \pm 0.0$ | $0.646935 \pm 0.0$ | $5.3938 \pm 0.0$ |
| PNNS(KM++) | $6.75 \pm 0.07$ | $92.77290 \pm 0.00006$ | $0.456724 \pm 0.0$ | $0.646935 \pm 0.0$ | $5.3938 \pm 0.0$ |
| PNNS(GKM++) | $6.73782 \pm 0.00011$ | $92.77290 \pm 0.00006$ | $0.456724 \pm 0.0$ | $0.646935 \pm 0.0$ | $5.3938 \pm 0.0$ |
| PNNS(UNIF; $\rho = 10$) | $6.75 \pm 0.07$ | $92.77290 \pm 0.00006$ | $0.468 \pm 0.021$ | $0.82 \pm 0.39$ | $5.3938 \pm 0.0$ |
| PNNS(MAXMIN; $\rho = 10$) | $6.73786 \pm 0.00013$ | $92.77290 \pm 0.00006$ | $0.456724 \pm 0.0$ | $0.646935 \pm 0.0$ | $5.3938 \pm 0.0$ |
| PNNS(KM++; $\rho = 10$) | $6.73785 \pm 0.00012$ | $92.77290 \pm 0.00006$ | $0.456724 \pm 0.0$ | $0.646935 \pm 0.0$ | $5.3938 \pm 0.0$ |
| PNNS(GKM++; $\rho = 10$) | $6.73785 \pm 0.00012$ | $92.77290 \pm 0.00006$ | $0.456724 \pm 0.0$ | $0.646935 \pm 0.0$ | $5.3938 \pm 0.0$ |
| PNN | $6.73774 \pm 0.0$ | $92.7729$ | $0.456724 \pm 0.0$ | $0.646935$ | $5.3938$ |

### A.1.3 CI (average)

| seeder | A3 | Birch1 | Birch2 | Unbalance | Dim1024 |
|---|---|---|---|---|---|
| UNIF | $6.6 \pm 1.6$ | $6.7 \pm 1.7$ | $16.4 \pm 2.5$ | $3.93 \pm 0.35$ | $3.7 \pm 1.1$ |
| MAXMIN | $2.9 \pm 1.0$ | $5.6 \pm 1.3$ | $7.4 \pm 1.8$ | $0.9 \pm 0.6$ | $0$ |
| KM++ | $4.1 \pm 1.2$ | $4.8 \pm 1.2$ | $7.1 \pm 1.6$ | $0.5 \pm 0.6$ | $0.004 \pm 0.063$ |
| GKM++ | $1.7 \pm 0.8$ | $2.7 \pm 1.1$ | $1.6 \pm 0.8$ | $0.05 \pm 0.23$ | $0$ |
| REF(UNIF) | $4.7 \pm 1.1$ | $4.5 \pm 1.1$ | $12.4 \pm 1.8$ | $3.4 \pm 0.8$ | $1.4 \pm 0.7$ |
| REF(MAXMIN) | $0.8 \pm 0.5$ | $3.1 \pm 0.8$ | $2.4 \pm 1.0$ | $0$ | $0$ |
| REF(KM++) | $2.8 \pm 0.8$ | $3.3 \pm 1.0$ | $4.9 \pm 1.0$ | $0.00 \pm 0.04$ | $0$ |
| REF(GKM++) | $0.8 \pm 0.5$ | $1.6 \pm 0.7$ | $0.6 \pm 0.5$ | $0$ | $0$ |
| PNNS(UNIF) | $0.3 \pm 0.5$ | $0$ | $0.9 \pm 0.7$ | $0.4 \pm 0.6$ | $0.08 \pm 0.27$ |
| PNNS(MAXMIN) | $0$ | $0$ | $0$ | $0$ | $0$ |
| PNNS(KM++) | $0.02 \pm 0.14$ | $0$ | $0$ | $0$ | $0$ |
| PNNS(GKM++) | $0$ | $0$ | $0$ | $0$ | $0$ |
| PNNS(UNIF; $\rho = 10$) | $0.02 \pm 0.13$ | $0$ | $0.3 \pm 0.5$ | $0.2 \pm 0.4$ | $0$ |
| PNNS(MAXMIN; $\rho = 10$) | $0$ | $0$ | $0$ | $0$ | $0$ |
| PNNS(KM++; $\rho = 10$) | $0$ | $0$ | $0$ | $0$ | $0$ |
| PNNS(GKM++; $\rho = 10$) | $0$ | $0$ | $0$ | $0$ | $0$ |
| PNN | $0$ | $0$ | $0$ | $0$ | $0$ |

### A.1.4 Success rate

| seeder | A3 | Birch1 | Birch2 | Unbalance | Dim1024 |
|---|---|---|---|---|---|
| UNIF | 0 | 0 | 0 | 0 | 0.001 |
| MAXMIN | 0.004 | 0 | 0 | 0.223 | 1 |
| KM++ | 0 | 0 | 0 | 0.541 | 0.996 |
| GKM++ | 0.051 | 0.02 | 0.06 | 0.946 | 1 |
| REF(UNIF) | 0 | 0 | 0 | 0 | 0.055 |
| REF(MAXMIN) | 0.254 | 0 | 0.01 | 1 | 1 |
| REF(KM++) | 0.002 | 0 | 0 | 0.998 | 1 |
| REF(GKM++) | 0.239 | 0.03 | 0.44 | 1 | 1 |
| PNNS(UNIF) | 0.709 | 1 | 0.26 | 0.696 | 0.919 |
| PNNS(MAXMIN) | 1 | 1 | 1 | 1 | 1 |
| PNNS(KM++) | 0.98 | 1 | 1 | 1 | 1 |
| PNNS(GKM++) | 1 | 1 | 1 | 1 | 1 |
| PNNS(UNIF; $\rho = 10$) | 0.982 | 1 | 0.76 | 0.806 | 1 |
| PNNS(MAXMIN; $\rho = 10$) | 1 | 1 | 1 | 1 | 1 |
| PNNS(KM++; $\rho = 10$) | 1 | 1 | 1 | 1 | 1 |
| PNNS(GKM++; $\rho = 10$) | 1 | 1 | 1 | 1 | 1 |
| PNN | 1 | 1 | 1 | 1 | 1 |

## A.1.5 Normalized distance computations

| seeder | A3 | Birch1 | Birch2 | Unbalance | Dim1024 |
|---|---|---|---|---|---|
| UNIF | $1.80 \pm 0.18$ | $2.52 \pm 0.37$ | $1.146 \pm 0.027$ | $2.7 \pm 0.6$ | $1.89 \pm 0.32$ |
| MAXMIN | $1.33 \pm 0.09$ | $2.08 \pm 0.29$ | $1.073 \pm 0.020$ | $1.26 \pm 0.10$ | $1.07812 \pm 0.0$ |
| KM++ | $1.41 \pm 0.12$ | $2.00 \pm 0.25$ | $1.063 \pm 0.014$ | $1.35 \pm 0.31$ | $1.079 \pm 0.009$ |
| GKM++ | $5.08 \pm 0.05$ | $6.46 \pm 0.16$ | $5.972 \pm 0.004$ | $3.78 \pm 0.11$ | $3.89062 \pm 0.0$ |
| REF(UNIF) | $3.94 \pm 0.12$ | $3.75 \pm 0.21$ | $2.393 \pm 0.019$ | $4.6 \pm 0.5$ | $6.6 \pm 0.4$ |
| REF(MAXMIN) | $3.139 \pm 0.038$ | $3.08 \pm 0.11$ | $2.207 \pm 0.008$ | $2.468 \pm 0.019$ | $4.1136 \pm 0.0015$ |
| REF(KM++) | $3.39 \pm 0.07$ | $3.23 \pm 0.14$ | $2.240 \pm 0.012$ | $2.61 \pm 0.08$ | $4.1140 \pm 0.0023$ |
| REF(GKM++) | $6.992 \pm 0.029$ | $7.72 \pm 0.11$ | $7.1128 \pm 0.0030$ | $5.068 \pm 0.028$ | $6.9261 \pm 0.0014$ |
| PNNS(UNIF) | $3.84 \pm 0.06$ | $4.07 \pm 0.04$ | $3.191 \pm 0.012$ | $4.63 \pm 0.37$ | $4.27 \pm 0.16$ |
| PNNS(MAXMIN) | $3.283 \pm 0.025$ | $3.662 \pm 0.025$ | $3.044 \pm 0.007$ | $3.257 \pm 0.026$ | $3.380 \pm 0.027$ |
| PNNS(KM++) | $3.439 \pm 0.039$ | $3.791 \pm 0.031$ | $3.084 \pm 0.008$ | $3.40 \pm 0.05$ | $3.380 \pm 0.027$ |
| PNNS(GKM++) | $7.137 \pm 0.022$ | $8.416 \pm 0.020$ | $7.957 \pm 0.007$ | $5.903 \pm 0.034$ | $6.193 \pm 0.027$ |
| PNNS(UNIF; $\rho = 10$) | $12.37 \pm 0.10$ | $12.027 \pm 0.037$ | $11.314 \pm 0.035$ | $12.38 \pm 0.35$ | $20.6 \pm 0.5$ |
| PNNS(MAXMIN; $\rho = 10$) | $11.78 \pm 0.09$ | $11.567 \pm 0.030$ | $11.089 \pm 0.035$ | $11.32 \pm 0.13$ | $22.4 \pm 0.5$ |
| PNNS(KM++; $\rho = 10$) | $11.88 \pm 0.08$ | $11.716 \pm 0.037$ | $11.154 \pm 0.034$ | $11.44 \pm 0.13$ | $22.4 \pm 0.5$ |
| PNNS(GKM++; $\rho = 10$) | $15.63 \pm 0.08$ | $16.385 \pm 0.037$ | $16.015 \pm 0.032$ | $13.98 \pm 0.13$ | $25.2 \pm 0.5$ |
| PNN | $262.324$ | $1730.39$ | $1738.25$ | $1422.21$ | $531.759$ |

## A.1.6 Lloyd's iterations

| seeder | A3 | Birch1 | Birch2 | Unbalance | Dim1024 |
|---|---|---|---|---|---|
| UNIF | $26 \pm 10$ | $120 \pm 40$ | $45 \pm 12$ | $33 \pm 15$ | $2.7 \pm 1.3$ |
| MAXMIN | $19 \pm 10$ | $95 \pm 36$ | $41 \pm 11$ | $3.2 \pm 1.1$ | $1 \pm 0$ |
| KM++ | $21 \pm 9$ | $85 \pm 31$ | $41 \pm 13$ | $11 \pm 13$ | $1.00 \pm 0.07$ |
| GKM++ | $13 \pm 7$ | $60 \pm 23$ | $23 \pm 11$ | $3 \pm 6$ | $1 \pm 0$ |
| REF(UNIF) | $26 \pm 13$ | $100 \pm 40$ | $56 \pm 18$ | $32 \pm 20$ | $4.6 \pm 2.9$ |
| REF(MAXMIN) | $14 \pm 10$ | $68 \pm 34$ | $35 \pm 19$ | $2.3 \pm 0.6$ | $2.001 \pm 0.031$ |
| REF(KM++) | $21 \pm 11$ | $73 \pm 37$ | $46 \pm 19$ | $5 \pm 4$ | $2.001 \pm 0.031$ |
| REF(GKM++) | $12 \pm 8$ | $58 \pm 32$ | $17 \pm 15$ | $2.5 \pm 1.9$ | $2.000 \pm 0.019$ |
| PNNS(UNIF) | $18 \pm 7$ | $36 \pm 9$ | $38 \pm 19$ | $18 \pm 16$ | $3.1 \pm 1.0$ |
| PNNS(MAXMIN) | $10.3 \pm 3.4$ | $29 \pm 7$ | $12 \pm 4$ | $2.1 \pm 0.4$ | $2 \pm 0$ |
| PNNS(KM++) | $13 \pm 4$ | $31 \pm 8$ | $15 \pm 4$ | $5.4 \pm 4.0$ | $2 \pm 0$ |
| PNNS(GKM++) | $9.4 \pm 3.0$ | $25 \pm 6$ | $9.1 \pm 3.7$ | $2.9 \pm 2.2$ | $2 \pm 0$ |
| PNNS(UNIF; $\rho = 10$) | $11.2 \pm 3.2$ | $25 \pm 5$ | $18 \pm 13$ | $13 \pm 14$ | $3.0 \pm 0.4$ |
| PNNS(MAXMIN; $\rho = 10$) | $7.2 \pm 2.2$ | $20 \pm 4$ | $6.8 \pm 2.3$ | $2.5 \pm 1.2$ | $2.00 \pm 0.04$ |
| PNNS(KM++; $\rho = 10$) | $8.8 \pm 2.5$ | $22 \pm 4$ | $9.3 \pm 2.5$ | $4.0 \pm 2.2$ | $2.01 \pm 0.08$ |
| PNNS(GKM++; $\rho = 10$) | $6.8 \pm 2.1$ | $18.2 \pm 3.9$ | $5.9 \pm 2.0$ | $3.3 \pm 1.9$ | $2.000 \pm 0.015$ |
| PNN | $6$ | $10$ | $2$ | $1$ | $1$ |

## A.2 Real-world datasets (non-parallel)

We tested 100 samples for each dataset (30 for *UrbanGB*).

### A.2.1 Convergence time (in s)

| seeder | Bridge $(\times 10^{-2})$ | House $(\times 10^{-1})$ | Miss A. $(\times 10^{-1})$ | Urb.GB | Olivetti $(\times 10^{-2})$ | Isolet $(\times 10^{-1})$ |
|---|---|---|---|---|---|---|
| UNIF | $4.17 \pm 0.38$ | $3.3 \pm 0.6$ | $1.13 \pm 0.09$ | $1.63 \pm 0.23$ | $2.7 \pm 0.6$ | $1.52 \pm 0.30$ |
| MAXMIN | $4.6 \pm 0.5$ | $3.2 \pm 0.5$ | $1.78 \pm 0.33$ | $1.73 \pm 0.20$ | $4.1 \pm 0.6$ | $1.58 \pm 0.21$ |
| KM++ | $4.22 \pm 0.36$ | $2.40 \pm 0.37$ | $1.15 \pm 0.11$ | $1.56 \pm 0.13$ | $4.10 \pm 0.39$ | $1.75 \pm 0.29$ |
| GKM++ | $6.45 \pm 0.35$ | $3.33 \pm 0.33$ | $1.58 \pm 0.13$ | $5.51 \pm 0.16$ | $8.8 \pm 0.6$ | $3.02 \pm 0.29$ |
| REF(UNIF) | $22.0 \pm 0.5$ | $6.5 \pm 0.4$ | $3.26 \pm 0.11$ | $3.65 \pm 0.28$ | $8.0 \pm 0.5$ | $1.68 \pm 0.24$ |
| REF(MAXMIN) | $21.3 \pm 0.6$ | $6.7 \pm 0.5$ | $3.49 \pm 0.17$ | $3.26 \pm 0.24$ | $8.8 \pm 0.5$ | $1.56 \pm 0.22$ |
| REF(KM++) | $21.5 \pm 0.5$ | $5.76 \pm 0.38$ | $3.35 \pm 0.12$ | $3.13 \pm 0.20$ | $9.1 \pm 0.4$ | $1.80 \pm 0.23$ |
| REF(GKM++) | $24.6 \pm 0.6$ | $6.48 \pm 0.36$ | $3.99 \pm 0.19$ | $5.36 \pm 0.23$ | $12.0 \pm 0.4$ | $2.19 \pm 0.22$ |
| PNNS(UNIF) | $5.90 \pm 0.31$ | $4.74 \pm 0.31$ | $1.52 \pm 0.09$ | $4.12 \pm 0.17$ | $6.2 \pm 1.0$ | $1.63 \pm 0.16$ |
| PNNS(MAXMIN) | $5.88 \pm 0.29$ | $4.94 \pm 0.36$ | $1.68 \pm 0.14$ | $4.33 \pm 0.18$ | $6.6 \pm 1.4$ | $1.79 \pm 0.25$ |
| PNNS(KM++) | $5.9 \pm 0.4$ | $4.44 \pm 0.31$ | $1.52 \pm 0.10$ | $3.58 \pm 0.16$ | $6.74 \pm 0.38$ | $1.66 \pm 0.17$ |
| PNNS(GKM++) | $8.13 \pm 0.39$ | $5.22 \pm 0.32$ | $1.89 \pm 0.11$ | $5.99 \pm 0.14$ | $9.51 \pm 0.39$ | $2.07 \pm 0.18$ |
| PNNS(UNIF; $\rho = 10$) | $15.8 \pm 0.9$ | $8.88 \pm 0.32$ | $3.31 \pm 0.08$ | $10.76 \pm 0.15$ | $21.1 \pm 1.0$ | $3.85 \pm 0.21$ |
| PNNS(MAXMIN; $\rho = 10$) | $14.5 \pm 0.8$ | $8.67 \pm 0.34$ | $2.95 \pm 0.12$ | $10.84 \pm 0.20$ | $21.5 \pm 0.7$ | $3.71 \pm 0.22$ |
| PNNS(KM++; $\rho = 10$) | $15.2 \pm 0.8$ | $8.33 \pm 0.30$ | $3.17 \pm 0.10$ | $9.97 \pm 0.18$ | $21.8 \pm 0.5$ | $3.87 \pm 0.20$ |
| PNNS(GKM++; $\rho = 10$) | $17.9 \pm 0.9$ | $9.4 \pm 0.4$ | $3.47 \pm 0.11$ | $12.04 \pm 0.15$ | $25.0 \pm 0.7$ | $4.20 \pm 0.17$ |
| PNN | $36.2 \pm 1.2$ | $93.68 \pm 0.20$ | $9.356 \pm 0.033$ | $819 \pm 7$ | $30.5 \pm 1.8$ | $297.2 \pm 2.8$ |

### A.2.2 SSE cost (average)

| seeder | Bridge $(\times 10^{7})$ | House $(\times 10^{5})$ | Miss A. $(\times 10^{5})$ | Urb.GB $(\times 10^{2})$ | Olivetti $(\times 10^{4})$ | Isolet $(\times 10^{5})$ |
|---|---|---|---|---|---|---|
| UNIF | $1.178 \pm 0.009$ | $10.11 \pm 0.13$ | $6.07 \pm 0.05$ | $6.8 \pm 1.1$ | $1.296 \pm 0.027$ | $1.196 \pm 0.010$ |
| MAXMIN | $1.139 \pm 0.005$ | $10.18 \pm 0.07$ | $5.794 \pm 0.028$ | $2.99 \pm 0.07$ | $1.252 \pm 0.015$ | $1.233 \pm 0.014$ |
| KM++ | $1.154 \pm 0.007$ | $9.60 \pm 0.04$ | $5.687 \pm 0.037$ | $2.72 \pm 0.06$ | $1.276 \pm 0.021$ | $1.196 \pm 0.010$ |
| GKM++ | $1.124 \pm 0.004$ | $9.529 \pm 0.027$ | $5.507 \pm 0.016$ | $2.430 \pm 0.020$ | $1.227 \pm 0.013$ | $1.190 \pm 0.007$ |
| REF(UNIF) | $1.162 \pm 0.006$ | $9.95 \pm 0.10$ | $5.850 \pm 0.035$ | $5.3 \pm 0.4$ | $1.298 \pm 0.023$ | $1.189 \pm 0.006$ |
| REF(MAXMIN) | $1.160 \pm 0.005$ | $9.96 \pm 0.07$ | $5.845 \pm 0.033$ | $2.94 \pm 0.05$ | $1.258 \pm 0.020$ | $1.234 \pm 0.013$ |
| REF(KM++) | $1.158 \pm 0.005$ | $9.592 \pm 0.038$ | $5.792 \pm 0.033$ | $2.63 \pm 0.04$ | $1.283 \pm 0.022$ | $1.191 \pm 0.008$ |
| REF(GKM++) | $1.159 \pm 0.005$ | $9.562 \pm 0.027$ | $5.826 \pm 0.031$ | $2.420 \pm 0.022$ | $1.255 \pm 0.020$ | $1.1837 \pm 0.0036$ |
| PNNS(UNIF) | $1.124 \pm 0.005$ | $9.553 \pm 0.034$ | $5.607 \pm 0.040$ | $2.70 \pm 0.06$ | $1.215 \pm 0.014$ | $1.1800 \pm 0.0019$ |
| PNNS(MAXMIN) | $1.0947 \pm 0.0027$ | $9.508 \pm 0.025$ | $5.339 \pm 0.008$ | $2.388 \pm 0.013$ | $1.192 \pm 0.008$ | $1.1844 \pm 0.0029$ |
| PNNS(KM++) | $1.1076 \pm 0.0033$ | $9.486 \pm 0.020$ | $5.403 \pm 0.013$ | $2.323 \pm 0.010$ | $1.209 \pm 0.013$ | $1.1795 \pm 0.0016$ |
| PNNS(GKM++) | $1.0947 \pm 0.0029$ | $9.476 \pm 0.020$ | $5.342 \pm 0.010$ | $2.297 \pm 0.004$ | $1.189 \pm 0.007$ | $1.1790 \pm 0.0015$ |
| PNNS(UNIF; $\rho = 10$) | $1.1020 \pm 0.0032$ | $9.526 \pm 0.030$ | $5.447 \pm 0.022$ | $2.604 \pm 0.032$ | $1.181 \pm 0.008$ | $1.1807 \pm 0.0020$ |
| PNNS(MAXMIN; $\rho = 10$) | $1.0861 \pm 0.0022$ | $9.482 \pm 0.021$ | $5.327 \pm 0.007$ | $2.349 \pm 0.010$ | $1.1669 \pm 0.0034$ | $1.1849 \pm 0.0025$ |
| PNNS(KM++; $\rho = 10$) | $1.0884 \pm 0.0026$ | $9.486 \pm 0.023$ | $5.336 \pm 0.008$ | $2.311 \pm 0.008$ | $1.174 \pm 0.006$ | $1.1806 \pm 0.0020$ |
| PNNS(GKM++; $\rho = 10$) | $1.0850 \pm 0.0020$ | $9.474 \pm 0.022$ | $5.325 \pm 0.008$ | $2.2992 \pm 0.0033$ | $1.1671 \pm 0.0032$ | $1.1795 \pm 0.0018$ |
| PNN | $1.08279$ | $9.49701$ | $5.31588$ | $2.3153$ | $1.16238$ | $1.17692$ |

### A.2.3 SSE cost (minimum)

| seeder | $Bridge$ $(\times 10^7)$ | $House$ $(\times 10^5)$ | $Miss\ A.$ $(\times 10^5)$ | $Urb.GB$ $(\times 10^2)$ | $Olivetti$ $(\times 10^4)$ | $Isolet$ $(\times 10^5)$ |
|---|---|---|---|---|---|---|
| UNIF | 1.15723 | 9.87713 | 5.95536 | 5.42031 | 1.23747 | 1.1795 |
| MAXMIN | 1.12869 | 10.0159 | 5.72134 | 2.90322 | 1.21735 | 1.20639 |
| KM++ | 1.14219 | 9.52638 | 5.59949 | 2.60788 | 1.23035 | 1.18025 |
| GKM++ | 1.11421 | 9.45992 | 5.47583 | 2.37869 | 1.19908 | 1.17709 |
| REF(UNIF) | 1.14562 | 9.77128 | 5.76604 | 4.39285 | 1.24007 | 1.17688 |
| REF(MAXMIN) | 1.14719 | 9.81462 | 5.7451 | 2.84408 | 1.22378 | 1.20994 |
| REF(KM++) | 1.13947 | 9.50834 | 5.72392 | 2.5686 | 1.22014 | 1.17933 |
| REF(GKM++) | 1.14344 | 9.49776 | 5.74112 | 2.37821 | 1.2116 | 1.17737 |
| PNNS(UNIF) | 1.11345 | 9.47461 | 5.51748 | 2.5921 | 1.17155 | 1.17685 |
| PNNS(MAXMIN) | 1.08926 | 9.45145 | 5.32153 | 2.368 | 1.17443 | 1.17977 |
| PNNS(KM++) | 1.10041 | 9.44324 | 5.37209 | 2.30731 | 1.18579 | 1.17687 |
| PNNS(GKM++) | 1.08729 | 9.42885 | 5.31768 | 2.28822 | 1.17066 | 1.17695 |
| PNNS(UNIF; $\rho = 10$) | 1.09472 | 9.45075 | 5.39731 | 2.51072 | 1.16219 | 1.17689 |
| PNNS(MAXMIN; $\rho = 10$) | 1.07998 | 9.43216 | 5.30742 | 2.33416 | 1.15955 | 1.18024 |
| PNNS(KM++; $\rho = 10$) | 1.08029 | 9.42166 | 5.31398 | 2.29434 | 1.16141 | 1.17694 |
| PNNS(GKM++; $\rho = 10$) | 1.08122 | 9.41836 | 5.30621 | 2.29477 | 1.16036 | 1.17689 |
| PNN | 1.08279 | 9.49701 | 5.31588 | 2.3153 | 1.16238 | 1.17692 |

### A.2.4 Normalized distance computations

| seeder | $Bridge$ | $House$ | $Miss\ A.$ | $Urb.GB$ | $Olivetti$ | $Isolet$ |
|---|---|---|---|---|---|---|
| UNIF | $8.4 \pm 0.5$ | $5.7 \pm 0.9$ | $13.5 \pm 0.9$ | $1.60 \pm 0.23$ | $1.79 \pm 0.14$ | $3.35 \pm 0.36$ |
| MAXMIN | $8.9 \pm 0.8$ | $4.5 \pm 0.6$ | $23 \pm 4$ | $1.196 \pm 0.037$ | $2.21 \pm 0.07$ | $3.14 \pm 0.25$ |
| KM++ | $8.2 \pm 0.6$ | $3.71 \pm 0.39$ | $13.7 \pm 1.3$ | $1.229 \pm 0.029$ | $2.26 \pm 0.07$ | $3.36 \pm 0.35$ |
| GKM++ | $13.2 \pm 0.5$ | $9.27 \pm 0.32$ | $19.5 \pm 1.5$ | $8.168 \pm 0.033$ | $6.04 \pm 0.05$ | $6.92 \pm 0.33$ |
| REF(UNIF) | $44.8 \pm 0.8$ | $8.3 \pm 0.6$ | $39.4 \pm 1.0$ | $3.15 \pm 0.15$ | $5.68 \pm 0.10$ | $4.62 \pm 0.26$ |
| REF(MAXMIN) | $43.5 \pm 0.8$ | $7.4 \pm 0.5$ | $42.2 \pm 1.9$ | $2.509 \pm 0.029$ | $6.40 \pm 0.08$ | $4.48 \pm 0.21$ |
| REF(KM++) | $43.8 \pm 0.7$ | $6.78 \pm 0.39$ | $40.5 \pm 1.3$ | $2.62 \pm 0.08$ | $6.53 \pm 0.10$ | $4.75 \pm 0.25$ |
| REF(GKM++) | $49.5 \pm 0.8$ | $12.25 \pm 0.35$ | $48.3 \pm 2.1$ | $9.465 \pm 0.021$ | $10.34 \pm 0.07$ | $8.20 \pm 0.22$ |
| PNNS(UNIF) | $10.81 \pm 0.37$ | $6.80 \pm 0.36$ | $16.6 \pm 0.9$ | $4.03 \pm 0.09$ | $4.25 \pm 0.10$ | $5.06 \pm 0.15$ |
| PNNS(MAXMIN) | $10.82 \pm 0.37$ | $6.11 \pm 0.31$ | $19.2 \pm 1.5$ | $3.335 \pm 0.025$ | $4.82 \pm 0.07$ | $5.27 \pm 0.24$ |
| PNNS(KM++) | $10.6 \pm 0.4$ | $6.05 \pm 0.29$ | $16.7 \pm 1.0$ | $3.31 \pm 0.06$ | $5.00 \pm 0.10$ | $5.24 \pm 0.17$ |
| PNNS(GKM++) | $15.9 \pm 0.4$ | $11.59 \pm 0.30$ | $22.2 \pm 1.2$ | $10.191 \pm 0.014$ | $8.76 \pm 0.09$ | $8.73 \pm 0.16$ |
| PNNS(UNIF; $\rho = 10$) | $22.3 \pm 0.4$ | $15.20 \pm 0.34$ | $32.2 \pm 0.9$ | $12.21 \pm 0.09$ | $14.77 \pm 0.22$ | $16.26 \pm 0.31$ |
| PNNS(MAXMIN; $\rho = 10$) | $20.00 \pm 0.35$ | $14.25 \pm 0.32$ | $28.9 \pm 1.4$ | $11.301 \pm 0.032$ | $15.48 \pm 0.19$ | $15.46 \pm 0.26$ |
| PNNS(KM++; $\rho = 10$) | $21.11 \pm 0.37$ | $14.35 \pm 0.29$ | $31.1 \pm 1.2$ | $11.298 \pm 0.026$ | $15.64 \pm 0.22$ | $16.38 \pm 0.30$ |
| PNNS(GKM++; $\rho = 10$) | $26.18 \pm 0.32$ | $19.98 \pm 0.33$ | $35.4 \pm 1.2$ | $18.139 \pm 0.021$ | $19.38 \pm 0.20$ | $19.52 \pm 0.25$ |
| PNN | 51.8177 | 271.383 | 87.8464 | 1351 | 19.7431 | 739.388 |

### A.2.5 Lloyd's iterations

| seeder | Bridge | House | Miss A. | Urb.GB | Olivetti | Isolet |
|---|---|---|---|---|---|---|
| UNIF | $21 \pm 5$ | $124 \pm 29$ | $31 \pm 7$ | $120 \pm 28$ | $8.1 \pm 2.6$ | $38 \pm 14$ |
| MAXMIN | $25 \pm 6$ | $134 \pm 35$ | $62 \pm 17$ | $96 \pm 29$ | $8.1 \pm 2.6$ | $28 \pm 9$ |
| KM++ | $22 \pm 5$ | $100 \pm 26$ | $36 \pm 7$ | $84 \pm 17$ | $8.1 \pm 2.4$ | $38 \pm 14$ |
| GKM++ | $20 \pm 4$ | $90 \pm 25$ | $37 \pm 9$ | $86 \pm 25$ | $6.1 \pm 1.5$ | $34 \pm 12$ |
| REF(UNIF) | $26 \pm 10$ | $116 \pm 27$ | $40 \pm 13$ | $130 \pm 60$ | $10 \pm 4$ | $37 \pm 17$ |
| REF(MAXMIN) | $27 \pm 11$ | $140 \pm 40$ | $53 \pm 18$ | $100 \pm 60$ | $9 \pm 4$ | $30 \pm 15$ |
| REF(KM++) | $25 \pm 10$ | $104 \pm 30$ | $44 \pm 15$ | $110 \pm 50$ | $9 \pm 4$ | $36 \pm 17$ |
| REF(GKM++) | $27 \pm 12$ | $98 \pm 29$ | $52 \pm 20$ | $100 \pm 50$ | $8 \pm 4$ | $33 \pm 16$ |
| PNNS(UNIF) | $24 \pm 6$ | $99 \pm 24$ | $38 \pm 9$ | $109 \pm 32$ | $7.8 \pm 2.6$ | $33 \pm 13$ |
| PNNS(MAXMIN) | $28 \pm 6$ | $114 \pm 30$ | $53 \pm 15$ | $140 \pm 40$ | $7.2 \pm 2.6$ | $40 \pm 18$ |
| PNNS(KM++) | $25 \pm 6$ | $98 \pm 27$ | $41 \pm 10$ | $93 \pm 30$ | $7.6 \pm 2.6$ | $34 \pm 14$ |
| PNNS(GKM++) | $24 \pm 7$ | $93 \pm 26$ | $42 \pm 11$ | $86 \pm 28$ | $5.6 \pm 2.3$ | $30 \pm 14$ |
| PNNS(UNIF; $\rho = 10$) | $18 \pm 5$ | $90 \pm 24$ | $32 \pm 7$ | $103 \pm 30$ | $4.2 \pm 1.5$ | $32 \pm 13$ |
| PNNS(MAXMIN; $\rho = 10$) | $17 \pm 4$ | $93 \pm 27$ | $36 \pm 10$ | $111 \pm 38$ | $3.5 \pm 1.1$ | $32 \pm 14$ |
| PNNS(KM++; $\rho = 10$) | $17 \pm 5$ | $85 \pm 23$ | $34 \pm 9$ | $86 \pm 34$ | $3.9 \pm 1.2$ | $29 \pm 12$ |
| PNNS(GKM++; $\rho = 10$) | $16 \pm 5$ | $88 \pm 26$ | $33 \pm 9$ | $74 \pm 24$ | $3.7 \pm 1.0$ | $26 \pm 11$ |
| PNN | 13 | 78 | 26 | 98 | 5 | 14 |

### A.3   Real-world datasets (parallel, 4 threads)

We tested 30 samples for each dataset.

### A.3.1   Convergence time (in s)

| seeder | $Urb.GB$ | $USCensus$ |
|---|---|---|
| UNIF | $0.86 \pm 0.13$ | $13.2 \pm 3.3$ |
| MAXMIN | $1.06 \pm 0.12$ | $19.8 \pm 2.1$ |
| KM++ | $0.91 \pm 0.13$ | $17.8 \pm 2.2$ |
| GKM++ | $4.71 \pm 0.17$ | $42.5 \pm 3.0$ |
| REF(UNIF) | $1.74 \pm 0.12$ | $17.1 \pm 2.0$ |
| REF(MAXMIN) | $1.59 \pm 0.19$ | $23.7 \pm 2.1$ |
| REF(KM++) | $1.51 \pm 0.16$ | $22.3 \pm 2.2$ |
| REF(GKM++) | $2.34 \pm 0.13$ | $44.2 \pm 2.0$ |
| PNNS(UNIF) | $2.02 \pm 0.11$ | $16.0 \pm 1.0$ |
| PNNS(MAXMIN) | $2.11 \pm 0.17$ | $20.1 \pm 1.5$ |
| PNNS(KM++) | $1.78 \pm 0.10$ | $18.9 \pm 1.0$ |
| PNNS(GKM++) | $2.45 \pm 0.13$ | $34.8 \pm 1.1$ |
| PNNS(UNIF; $\rho = 10$) | $6.29 \pm 0.28$ | $50.2 \pm 1.2$ |
| PNNS(MAXMIN; $\rho = 10$) | $6.32 \pm 0.24$ | $53.1 \pm 1.6$ |
| PNNS(KM++; $\rho = 10$) | $6.04 \pm 0.24$ | $51.6 \pm 1.4$ |
| PNNS(GKM++; $\rho = 10$) | $6.38 \pm 0.18$ | $59.5 \pm 1.7$ |

### A.3.2   SSE cost (average)

| seeder | $Urb.GB$ $(\times 10^2)$ | $USCensus$ $(\times 10^7)$ |
|---|---|---|
| UNIF | $6.8 \pm 1.1$ | $1.200 \pm 0.010$ |
| MAXMIN | $2.99 \pm 0.07$ | $1.32 \pm 0.04$ |
| KM++ | $2.72 \pm 0.05$ | $1.192 \pm 0.011$ |
| GKM++ | $2.430 \pm 0.020$ | $1.172 \pm 0.007$ |
| REF(UNIF) | $5.26 \pm 0.35$ | $1.181 \pm 0.008$ |
| REF(MAXMIN) | $2.93 \pm 0.05$ | $1.300 \pm 0.032$ |
| REF(KM++) | $2.64 \pm 0.05$ | $1.180 \pm 0.006$ |
| REF(GKM++) | $2.413 \pm 0.017$ | $1.156 \pm 0.005$ |
| PNNS(UNIF) | $2.70 \pm 0.05$ | $1.1423 \pm 0.0014$ |
| PNNS(MAXMIN) | $2.382 \pm 0.013$ | $1.1760 \pm 0.0030$ |
| PNNS(KM++) | $2.325 \pm 0.008$ | $1.1434 \pm 0.0017$ |
| PNNS(GKM++) | $2.301 \pm 0.005$ | $1.1405 \pm 0.0007$ |
| PNNS(UNIF; $\rho = 10$) | $2.595 \pm 0.031$ | $1.1422 \pm 0.0014$ |
| PNNS(MAXMIN; $\rho = 10$) | $2.349 \pm 0.011$ | $1.1744 \pm 0.0031$ |
| PNNS(KM++; $\rho = 10$) | $2.311 \pm 0.006$ | $1.1432 \pm 0.0016$ |
| PNNS(GKM++; $\rho = 10$) | $2.297 \pm 0.004$ | $1.1402 \pm 0.0006$ |

### A.3.3   SSE cost (minimum)

| seeder | $Urb.GB$ $_{(\times 10^2)}$ | $USCensus$ $_{(\times 10^7)}$ |
|---|---|---|
| UNIF | 5.42031 | 1.18059 |
| MAXMIN | 2.90322 | 1.25624 |
| KM++ | 2.61092 | 1.17627 |
| GKM++ | 2.37869 | 1.15802 |
| REF(UNIF) | 4.65076 | 1.16404 |
| REF(MAXMIN) | 2.83593 | 1.2423 |
| REF(KM++) | 2.50803 | 1.16902 |
| REF(GKM++) | 2.37949 | 1.14828 |
| PNNS(UNIF) | 2.61665 | 1.14017 |
| PNNS(MAXMIN) | 2.36183 | 1.16818 |
| PNNS(KM++) | 2.31092 | 1.14155 |
| PNNS(GKM++) | 2.2858 | 1.13901 |
| PNNS(UNIF; $\rho = 10$) | 2.52545 | 1.1392 |
| PNNS(MAXMIN; $\rho = 10$) | 2.32724 | 1.16717 |
| PNNS(KM++; $\rho = 10$) | 2.3022 | 1.14102 |
| PNNS(GKM++; $\rho = 10$) | 2.2881 | 1.13892 |

