# OpenReview forum: "Systematically and efficiently improving $k$-means initialization by pairwise-nearest-neighbor smoothing"
_TMLR — Accepted by TMLR_

### Review · Reviewer_CQMT · 2022-08-11

**Summary Of Contributions:**

This paper studies the performance of PNN-based (PNN = pairwise nearest neighbor) init seeding method for k-means clustering. The PNN method is a meta method, in the sense that it needs to be combined with an existing seeding algorithm, and enhances its performance. In particular, the PNN method first partitions the entire dataset into J random parts (where J is a parameter), and then performs an existing method on each part to obtain kJ centers, and eventually greedily and repeatedly finds and merges the two parts whose merging leads to the smallest overall cost.

The focus of this paper is to compare the empirical performance of PNN, to many other popular methods, including the basic/non-meta methods such as uniform-sampling (called UNIF), MAXMIN, KM++, and another meta method, the refine method (called REF). In addition to comparing the running time and accuracy, the paper also proposes a hardware independent measure for the running time of clustering algorithms, the normalized distance computations (NDS), denoting the number of distance computations between two data points.

Extensive experiments have been done on both synthetic datasets and real datasets to evaluate the accuracy, convergence time for the follow-up Lloyd steps, and the NDC. For some larger datasets, parallel implementations are also provided and compared. Overall, the PNN methods work well compared with other baselines, and it almost always outperforms the corresponding basic seeding method without PNN, even though it requires a slightly more running time.

**Broader Impact Concerns:**

None.

**Requested Changes:**

Critical to securing my recommendation for acceptance:
----

1. In the beginning of section 3, you said the experiments have \rho = 1. Can you justify why this is the best choice? Perhaps you will need additional experiments with different \rho’s, and also give a (theoretical) argument/explanation why this should happen. Moreover, it would be interesting to explore how this parameter should be set, for different types of datasets and init seeding methods?

2. You pick different accelerator methods for every dataset. Does the choice of accelerator potentially the relative merit of PNN over other methods? If so, shouldn’t one use the same accelerator (or none), for fair comparison?

Changes that strengthen your work:
----

1. Footnote in page 4: “triangular inequality” -> “triangle inequality”
2. It is suggested to see how much percentage of time is spent on NDC, and how much is on other parts.
3. In the test of synthetic data, it is suggested to also list the actual costs that the algorithms achieve, since finding the solution optimally seems a goal that is too restrictive.
4. How do you implement the parallel algorithms? What part is parallelized?

**Strengths And Weaknesses:**

Strength:
---

Extensive experiments with many popular seeding methods are tested, and it is helpful for both practitioners and theorists to see how all of them compare. The proposal of NDC which is a hardware-independent measure for benchmarking the performance of clustering algorithms. This NDC measure is quite natural and indeed captures the “unit” of computation for clustering algorithms, which may be applied more widely and have a lasting impact.

Weakness:
---

While the effort of conducting the experiment and benchmarking is appreciated, I don’t see the significant claimed advantage of the PNN method. To me, it seems PNN performs very much similar to existing methods; while it achieves marginally better performance, it also uses more computations. Hence, it is not convincing that PNN has a clear advantage.

Another weakness is the lack of discussion/exploration of some potentially interesting experimental findings. For instance, it is claimed that \rho = 1 gives the best tradeoff - why is this? This is somewhat counterintuitive. Moreover, this paper does not really suggest new/improved methods, and hence lacks technical depth. For instance, regarding this PNN which is a meta method, is it useful to nest it, i.e., running another PNN as the basic clustering algorithm for each of the J parts?

---

> ### Author Response · Authors · 2022-09-15
> **Response part 1**
>
> > I don’t see the significant claimed advantage of the PNN method
>
> Please see our general considerations on this in the top-level thread.
>
> > Another weakness is the lack of discussion/exploration of some potentially interesting experimental findings. For instance, it is claimed that $\rho = 1$ gives the best tradeoff - why is this? This is somewhat counterintuitive.
>
> > In the beginning of section 3, you said the experiments have $\rho = 1$. Can you justify why this is the best choice? Perhaps you will need additional experiments with different $\rho$’s, and also give a (theoretical) argument/explanation why this should happen. Moreover, it would be interesting to explore how this parameter should be set, for different types of datasets and init seeding methods?
>
> We do not claim that $\rho = 1$ gives the *best* trade-off, we claim it is a good trade-off and a sensible default value. We have now strived to make this clearer in the revised version (page 6, point 1 after the algorithm), as follows:
>
>   "Throughout the main text we use the value $\rho=1$, which from our preliminary analysis seems to result in a good trade-off in all cases and for all INIT algorithms. This is a valid (i.e. non-degenerate) choice whenever $k<N/2$, which is arguably always the case in realistic scenarios, thus making it a reasonable default value."
>
> As for testing with different values of $\rho$, indeed we did that, and reported the results in the Appendix. Our experiments show that the SSE generally improves (although not in all cases) at the expense of increased computational time. The most interesting results are probably those in the tables A.2.1 and A.2.2. Our experiments in this direction did not suggest any "optimal" setting, even taking into account the dataset types, they only confirmed the general tendency that our analysis suggests: "as a rule of thumb, increasing $\rho$ improves quality at the cost of performance, although this is not strictly true in all cases." (p.6) The differences in SSE are also not very large (as expected from the fact that PNNS with $\rho=1$ already achieves results close to PNN, which would correspond to the large-$\rho$ limit). For the main text we opted to simplify the presentation and focus on a single generally good value of $\rho$.
>
> In the revised paper we have added these considerations to the "Discussion" section.
>
>
> > Moreover, this paper does not really suggest new/improved methods, and hence lacks technical depth. For instance, regarding this PNN which is a meta method, is it useful to nest it, i.e., running another PNN as the basic clustering algorithm for each of the J parts?
>
> Indeed, we experimented with that idea. As reported in the discussion, "Another option would be to apply the PNN-smoothing scheme recursively (e.g. PNNS(PNNS(KM++)) and similar): we thoroughly explored this possibility, but concluded that it does not bring significant advantages over simply using a larger $\rho$." The attached code has an `rlevel` setting for PNNS. We have also tested a fully-recursive version, which in the code we call PNNSR, in which PNNS is applied recursively until $N < 2k$, at which point it uses PNN. The results are indeed rather good, but at the cost of convergence time/NDCs (the fully-recursive version has a complexity that scales at least like $N \log N$), and the trade-off is not clearly better than that of varying $\rho$, as reported; in presenting our results, we opted for simplicity. We have now added a footnote mentioning PNNSR.

---

> ### Author Response · Authors · 2022-09-15
> **Response part 2**
>
> > You pick different accelerator methods for every dataset. Does the choice of accelerator potentially the relative merit of PNN over other methods? If so, shouldn’t one use the same accelerator (or none), for fair comparison?
>
> Using the same accelerator for all tests was actually our first setup. The results don't change much, as we reported in the discussion: "We also verified that, as one would expect, the results are qualitatively the same regardless of the scheme used to accelerate k-means iteration." The choice to optimize the accelerator is intended to showcase the best possible results that can be achieved with state-of-the-art methods, and it actually puts PNNS(INIT) at a disadvantage compared to INIT, since accelerating Lloyd's iteration reduces its advantage. The reason is that, as a general tendency, all accelerators become more and more effective as the iterations progress. In PNNS, we have to pay the price of starting up the iterations twice (this is captured by the NDC metric and discussed in the text, see p.6 point 5). Thus, by using the optimal accelerator, one of the advantages of PNNS, namely that the final optimization is shorter (because we start closer to a local optimum), is diminished, even if the overall number of Lloyd's iterations is comparable (see table A.2.6). In conclusion, using a sub-optimal accelerator would only result in reducing the time gap between INIT and PNNS(INIT).
>
> In the revised version, we added these clarifications in section 4.1 ("Experimental setup").
>
> > Footnote in page 4: “triangular inequality” -> “triangle inequality”
>
> Fixed, thank you.
>
> > It is suggested to see how much percentage of time is spent on NDC, and how much is on other parts.
>
> We thank the reviewer for the suggestion. This is indeed an interesting point.
>
> Unfortunately, this is really difficult to assess reliably, since measuring the times spent in various parts of the algorithm affects the timing themselves. For instance, for the smallest datasets (A3, Unbalance, Dim1024) we had to run all simulations twice, once with logging activated and once with logging off, in order to measure the timings precisely. Distance computations are particularly sensitive to this, since their speed is greatly affected by caching and other mechanisms that may get disrupted by inserting extra instructions (depending on the circumstance, distances may be computed in bulk with BLAS routines, or in bulk with SIMD instructions, or on a one-by-one basis).
>
> Using a sampling profiler (as we did during development, to guide our implementation strategies) only provides a rough idea of the relative weight of distance computations.
>
> In conclusion, this depends in a very complicated way on the characteristics of the dataset, the accelerator used, and the hardware. A careful numerical analysis capable of disentangling all these effects would be of interest independently of PNNS, and it would require a dedicated effort.
>
> > In the test of synthetic data, it is suggested to also list the actual costs that the algorithms achieve, since finding the solution optimally seems a goal that is too restrictive.
>
> The costs are reported in the Appendix, see table A.1.2. In the main text, we only report the success rate because we believe that finding the optimal solution for synthetic datasets is not too restrictive: it is actually the reason for which those datasets were built and tested in the first place, and it allows a direct comparison with the existing literature (Fränti & Sieranoja 2019). Furthermore, PNNS is able to solve all the datasets in basically 100% of the cases, demonstrating that the goal is not too ambitious.
>
> Still, in order to provide a less restrictive metric that is still more interpretable than the SSE cost, in the revised version of the paper we have included a new table in the Appendix (A.1.3) where we also report the average CI values (the CI can be interpreted as the average number of mistakes).
>
> > How do you implement the parallel algorithms? What part is parallelized?
>
> In the Lloyd's iterations, we parallelize over the $N$ data points, as usually done. In the PNN procedure we parallelize over $\hat{k}$ (as explained in more detail in p.4). We have now added this information in the "Experimental setup" section 4.1.

---

### Review · Reviewer_5asV · 2022-08-25

**Summary Of Contributions:**

This paper proposes a meta-method for the initialization of the $k$-means clustering. The method first splits the dataset into $J$ random subsets and clusters them separately, then merges the $Jk$ clusters to $k$ clusters by applying the pairwise-nearest-neighbor method to the $Jk$ centroids. The proposed initialization method can be readily combined with any seeding algorithm (followed by Lloyd's algorithm), whose computational complexity is linear in $k$ and the data size $N$, that is performed on the $J$ subsets. Extensive numerical experiments on both synthetic and real datasets have been performed to demonstrate systematically superior results with respect to the similar "refine" smoothing scheme.

**Broader Impact Concerns:**

Not applicable.

**Requested Changes:**

See the Weaknesses above.

**Strengths And Weaknesses:**

Strengths: The idea is simple and the authors have performed sufficient numerical experiments.

Weaknesses:

1. According to the numerical results, the advantages of the method proposed in this work over the similar "refine" smoothing scheme seem to be minor for most cases (except for the case of convergence time for real-world datasets).

2. There is no theoretical justification for the proposed method and some mathematical notions are vague. For example,
* What is the precise definition of "the minimum SSE cost" (first appears on Page 10, Table 3)?

* For Lloyd's algorithm, the set of centroids $\mathcal{C}$ and the partition of data points are mutually dependent. I suggest that in (1), the sum-of-squared errors (SSE) should be defined only with respect to $\mathcal{C}$ (or $\mathcal{P}$), instead of both $\mathcal{C}$ and $\mathcal{P}$.

* The sentence "We thus define the number of normalized distance computations (NDC) as the total number of distance computations between $D$-dimensional vectors, divided by $Nk$" (Page 2) is confusing to me.  I hope that the authors can provide a precise mathematical definition.

3. The whole procedure (Algorithm 1) is used as an initialization method for the $k$-means clustering, but in the subsets, Lloyd's iterations are performed (which is the whole clustering procedure, rather than just initialization). This is a bit weird for me, especially when the number of subsets $J$ is pretty small (e.g., according to Table 1, if setting $\rho =1$, for the Bridge and Olivetti datasets, $J = 3$).

---

> ### Author Response · Authors · 2022-09-15
> **Response part 1**
>
> > According to the numerical results, the advantages of the method proposed in this work over the similar "refine" smoothing scheme seem to be minor for most cases (except for the case of convergence time for real-world datasets).
>
> We respectfully disagree with the reviewer on this assessment. We wrote a general comment in a top-level thread. Here we provide a more specific answer about the comparison with "refine" as well.
>
> In the synthetic datasets case, the original method by Bradley and Fayyad, REF(UNIF), has very poor performance in all datasets. The best variant, REF(GKM++), has very poor performance in the A3, Birch1, Birch2 cases, which are the most difficult. In contrast, PNNS(INIT) easily gets basically 100% success rate in all of them (even PNNS(UNIF) would only require a few restarts to solve all the problems). See table 2.
>
> In the real-world tests, as we report "PNNS(INIT) consistently achieves better costs than both INIT and REF(INIT), for all datasets and all the tested INIT." and also "In many cases, the average time for PNNS(INIT) is shorter than for REF(INIT)" (page 12). Looking at figure 2, one can clearly see that, if anything, REF(INIT) provides minor advantages compared to INIT in terms of SSE, at a much more significant expense in computational time (with few exceptions).
>
> > There is no theoretical justification for the proposed method
>
> We agree with the reviewer that having a detailed theoretical justification would be extremely useful, interesting and potentially fruitful. The difficulty in devising one is not surprising in this context, however: of all the tested methods, only KM++ has a theoretical justification (at least so far) as far as we are aware. For KM++ Arthur & Vassilvitskii (2007) exhibit an upper bound for the seeding process, which however does not take into account the following local optimization (and is thus quite loose). Since PNNS includes local optimization in the seeding process itself, one can see how the difficulty is exacerbated further.
>
> Still, we believe that the fact that, as noted in page 6 of the text, PNNS(INIT) interpolates between INIT and PNN as $\rho$ increases is a relevant insight and provides some intuitive justification for its general effectiveness.
>
> > and some mathematical notions are vague. For example, What is the precise definition of "the minimum SSE cost" (first appears on Page 10, Table 3)?
>
> The minimum SSE cost is just the lowest value of the SSE that was achieved in any of the runs on a given dataset with a given algorithm. This metric is commonly reported in similar works, e.g. Arthur & Vassilvitskii 2007, Celebi et al 2013 and others. It is useful to show what one could achieve with the popular repeated-kmeans (or multi-start-kmeans) strategy of running several independent stochastic optimizations and keeping the best value.
>
> In the revised version, we added an explanation about this in the text, section 4.3 (page 12).

---

> ### Author Response · Authors · 2022-09-15
> **Response part 2**
>
> > For Lloyd's algorithm, the set of centroids $\mathcal{C}$ and the partition of data points are mutually dependent. I suggest that in (1), the sum-of-squared errors (SSE) should be defined only with respect to $\mathcal{C}$ (or $\mathcal{P}$), instead of both $\mathcal{C}$ and $\mathcal{P}$.
>
> In the literature, all three options for the definition of the SSE ($\mathcal{C}$-only, $\mathcal{P}$-only, both) are commonly found, depending on personal preference and on context. (Actually some authors don't even bother providing a definition since the problem is so well-known.) For example, Bachem et al. (2016) use the more explicit one that depends both $\mathcal{P}$ and $\mathcal{C}$. In our work we chose to do the same, because we believe that it makes the remainder of the text clearer.
>
> > The sentence "We thus define the number of normalized distance computations (NDC) as the total number of distance computations between $D$-dimensional vectors, divided by $Nk$" (Page 2) is confusing to me. I hope that the authors can provide a precise mathematical definition.
>
> The definition is just this: whenever at any point in an algorithm we need to compute a distance (or a squared distance) between two $D$-dimensional vectors, we count one distance computation. At the end, we divide the total count by $Nk$.
> In the revised version, we have expanded the text to make it clearer.
>
> > The whole procedure (Algorithm 1) is used as an initialization method for the $k$-means clustering, but in the subsets, Lloyd's iterations are performed (which is the whole clustering procedure, rather than just initialization). This is a bit weird for me, especially when the number of subsets $J$ is pretty small (e.g., according to Table 1, if setting $\rho =1$, for the Bridge and Olivetti datasets, $J = 3$).
>
> The "refine" method also uses clustering as part of the seeding, so it is unclear to us why this would be weird. It is also unclear to us why using $J=3$ would be problematic: as noted in the text (page 6, point 1) the algorithm makes sense as long as $1 < J \le N/k$, and in fact the results for Bridge and Olivetti are good.

---

### Review · Reviewer_eetN · 2022-09-13

**Summary Of Contributions:**

This paper presents a meta-method for initializing the k-means clustering algorithm called PNN-smoothing. It consists in splitting a given dataset into J random subsets, clustering each of them individually, and merging the clustering results with the pairwise-nearest-neighbor (PNN) method.

**Requested Changes:**

Please see *Strengths And Weaknesses

**Strengths And Weaknesses:**

Strength

The proposed method is simple and easy to understand.


Weakness

1. This paper propose a strategy or trick (rather than an algorithm) to improve $k$-means initialization. Insides, it employs existing clustering algorithms, such as kmeans++, and existing merging method PNN. So the strategy is more like a combination and the contribution is limitted.

2. The used strategy is easy and straight and can be easily devoloped with a low threshold.

3. The clustering methods are not evaluated properly in experiment. There only one performance metric, i.e. success rate. Also success rates are not reported on real-world data (the results on synthetic data cannot be used in evaluation. They are only a showcase of your motivations etc.). In addition, the definition of success rate is not introduced properly.

4. The proposed strategy shows worse results on multiple metrics, such as convergence time and normalized distance computations.

5. The authors are expected to compare the complexiy of all considered methods with the proposed one. Only comparing the runtime is not enough, since there are so many factors can affect the runtime, such as the code quality, the programming language, running evironment, etc.

---

> ### Author Response · Authors · 2022-09-15
> **Response**
>
> > This paper propose a strategy or trick (rather than an algorithm) to improve $k$-means initialization. Insides, it employs existing clustering algorithms, such as kmeans++, and existing merging method PNN. So the strategy is more like a combination and the contribution is limitted.
>
> We respectfully disagree with the reviewer on this, but this is besides the point: the [TMLR editorial policy](https://jmlr.org/tmlr/editorial-policies.html) explicitly states that "Papers should be accepted if they meet the criteria, even if the contribution or significance of the work is modest."
>
> > The clustering methods are not evaluated properly in experiment. There only one performance metric, i.e. success rate. Also success rates are not reported on real-world data
>
> The success rate is our main metric only for the synthetic datasets case (sec. 4.2), for which however we also report the SSE in the Appendix (table A.1.2). For these datasets, there is a clear ground-truth objective to be achieved, and thus it makes sense to measure the success rate at which algorithms are successful. It also allows a direct comparison with the literature, namely Fränti & Sieranoja (2019), where the same metric was introduced and used.
> In the revised version of the paper, we now also report the average CI (which, as explained in the text, can be interpreted as the number of mistakes when clustering these kind of data) in the Appendix (table A.1.3 of the revised version).
>
> For real world data using the success rate simply does not make sense, since in the vast majority of the cases there is no ground truth and the data points are not arranged in clearly defined isotropic Gaussian clusters. Our metrics in this case (mean and minimum SSE costs) are consistent with the existing literature on the subject (e.g. Arthur & Vassilvitskii (2007), Celebi et al. (2013), etc.).
>
> > (the results on synthetic data cannot be used in evaluation. They are only a showcase of your motivations etc.).
>
> We kindly ask the reviewer to clarify this point, as we don't understand what they mean here.
>
> > In addition, the definition of success rate is not introduced properly.
>
> The success rate is defined in page 8 as "the frequency with which [an algorithm] finds a solution, i.e. a configuration with CI = 0". The CI is defined immediately above that, after the reference to the work where it was originally introduced: "The CI is computed by matching each centroid of a clustering with its closest one from the ground truth, and counting the number of unmatched ground truth centroids."
>
> In the revised version of the paper, we have now accompanied this definition of the CI with an explicit formula.
>
> > The proposed strategy shows worse results on multiple metrics, such as convergence time and normalized distance computations.
>
> Convergence time and NDCs are obviously highly correlated, as the latter is intended as a hardware-independent proxy for the former. We respectfully disagree with the reviewer that PNNS simply shows "worse results". Please see our general considerations on this in the top-level thread.
>
> > The authors are expected to compare the complexiy of all considered methods with the proposed one. Only comparing the runtime is not enough, since there are so many factors can affect the runtime, such as the code quality, the programming language, running evironment, etc.
>
> We are not sure of what is the point that the reviewer is raising here, since we have certainly taken this aspect into consideration. First of all, we did indeed report the computational complexity of all the algorithms in the paper. Secondly, we introduced the hardware-independent NDC measure to very explicitly provide an alternative, hardware- and implementation-independent metric to runtime (Introduction, page 2), we provided estimates of the NDCs of each seeding algorithm (sections 2 and 3), and we measured the total NDCs in all our tests (tables 2, 3 in the main text and A.1.5, A.2.4 in the Appendix). On top of this, in an effort to remove any biases from implementation details, we checked that our implementations are competitive with the best open-source ones we could find (footnote 4, page 7); our code is attached as auxiliary material to this submission not only for the purpose of reproducibility, but also so that implementation details can be inspected and its performance independently checked. Again, measuring the running time is quite common in the related literature, due to its obvious practical relevance (and due to the difficulty in predicting the actual performance of algorithms on purely theoretical grounds).

---

### Author Response · Authors · 2022-09-15
**Revised version and General response to reviewers**

We thank the reviewers for their suggestions and comments. We have now uploaded a revised version that includes some additional data (table A.1.3, containing the average CI values obtained on synthetic datasets) and additional clarifications and discussions.

We respond in detail to the specific points raised by the reviewers in the respective threads. There is a common comment from all the reviewers however, about the usefulness of this method, that we thus address here.

Reviewer COMT wrote:

> I don’t see the significant claimed advantage of the PNN method

Reviewer 5asV wrote:

> According to the numerical results, the advantages of the method proposed in this work over the similar "refine" smoothing scheme seem to be minor for most cases (except for the case of convergence time for real-world datasets).

Reviewer eetN wrote:

> The proposed strategy shows worse results on multiple metrics, such as convergence time and normalized distance computations.

We respectfully disagree with these assessments. We offer these general considerations on this central topic:

1. In the cases when the structure of the data is well captured by the SSE objective (exemplified by the synthetic datasets), PNNS is the only seeding method that gives 100% success rate on all the tested datasets. This array of datasets was created to explicitly expose potential weaknesses of seeding methods, and PNNS is the only (linear or quasi-linear) method that exhibits none of those weaknesses. In a sense, this gives a degree of confidence that, when using PNNS, k-means will not get stuck into clear and easily-fixed mistakes, and that any variation in the optimization result is more likely to emerge from the features of the datasets not being aligned with the SSE objective.

2. As for real-world datasets: there is a trade-off in time vs performance when one considers an algorithm INIT with the corresponding PNNS(INIT): this is normal and expected from our analysis. However, the fact that PNNS(INIT) has systematically better costs is not trivial (note that REF(INIT) is not always better than INIT, and when it is the margins are rater small). In the comparison, PNNS(INIT) clearly outperforms REF(INIT) in terms of both costs and times. Also, consider GKM++: this is a version of KM++ which was explicitly devised to improve KM++ at the expense of additional computations. Comparing PNNS(KM++) with GKM++ we see that the former is nearly always a better way to improve the performance of KM++ in both cost (all datasets) and time (6/7 datasets). Note that GKM++ is used as a default in the popular scikit-learn package, and our results show that PNNS(KM++) would be a better default; this leads us to think that our method could have a significant impact if adopted. Finally, we note that PNNS has results of comparable quality with PNN, while being (almost) linear instead of quadratic in $N$: in other words, it achieves a significantly better trade-off.

---

### Decision · Action_Editors · 2022-12-02

**Recommendation:** Accept with minor revision

**Comment:**

The paper proposes a meta-algorithm for improving the initialization of k-means clustering. The idea is to split the data into subsets, cluster each subset, and then merge the resulting clusters using a pairwise-nearest neighbor algorithm.

### Reviewer recommendations

From the reviewers' final recommendations, two were (leaning) positive, while another remained negative. The main critiques raised by the reviewers were as follows.

(1) _Limited novelty_. Reviewers found the core ideas in this paper a modest departure from prior work, with the final solution combining existing ideas. From my reading, there appears to be some merit to this view. However, the TMLR guidelines suggest we do not index on technical novelty, provided the results are sufficiently convincing and interesting.

(2) _Lack of theoretical results_. Reviewers noted that the paper is purely empirical in nature, with no theoretical results. The authors agree with this in the response, noting that most prior works do not have such guarantees. I agree with the critique, and certainly some theory would strengthen the paper, but do not find it disqualifying, as empirical papers could still be interesting to the community.

(3) _Lack of clarity_. Some reviewers raised concerns about clarity of presentation, including of some mathematical notions. I agree that there is some scope for improved clarity, particularly in terms of take-home messages (see below).

(4) _Mixed empirical results_. All reviewers had questions about whether the proposed method shows enough empirical promise over existing methods. The author response noted that their method is the only one that yields 100% accuracy on synthetic tasks, while also yielding consistent improvements over the INIT baseline on real-world tasks. I tend to agree with the authors that the synthetic results are a useful sanity check that the proposed method helps avoid "clear and easily-fixed mistakes". On the real-world datasets, the authors argue that an increase in compute time over INIT might be acceptable if there is a consistent improvement in quality; and that consistent improvements over INIT are non-trivial, e.g., not being achieved by the REF method. I tend to agree with the authors' argument, though the consistent reviewer confusion on this point suggests scope to be much clearer in the discussion of results.

To summarize, while critiques (1)-(4) are reasonable, (1) and (2) do not strongly influence my decision per the TMLR guidelines. While I understand the arguments behind (4), on balance I tend to view the results as being of potential interest.


### AE recommendation

The TMLR guidelines for acceptance are based on the following two questions:

(a) _Are the claims made in the submission supported by accurate, convincing and clear evidence?_

(b) _Would some individuals in TMLR's audience be interested in the findings of this paper?_

For (a), the major potential issue is whether the empirical results indeed convey the potential benefits of the PNNS method. As above, I believe they do reasonably show the potential benefits of the method in settings where consistently improved SSE cost, with bounded increase in compute cost, is desirable.

For (b), two of the reviewers believe that readers may find this paper interesting. From my reading, I agree with this assessment.

Given these, we recommend acceptance with minor revisions as suggested below.


### Suggested revisions

- The title is a bit verbose. Perhaps "Systematically and efficiently improving k-means initialization by pairwise-nearest-neighbor smoothing" conveys the same effect.

- Section 2: it would be useful to have a table summarizing the complexity (+ possibly NDCs) of all methods listed.

- Section 2: for each method, the summary is provided in a single para. This ends up quite long for the last two methods, making it hard to parse. It would be better to split the paras, e.g., so that the discussion of computational complexity appears on a separate para.

- When referring to the refine method, e.g., "Our method starts out identically to refine", it might be useful to instead use "REF(INIT)", or use a different typeface for refine.

- Since all reviewers had questions about the significance of the empirical results, it would be useful to explicitly and succinctly state (e.g., above Section 4.1) what the goals of the experiments are: to highlight that the proposed method helps avoid "clear and easily-fixed mistakes" in synthetic settings, and offers consistently improved SSE cost with bounded increase in compute cost in real-world settings.

- Further to the above, the points on experiments raised in the "Revised version and General response to reviewers" should be incorporated in the discussion of the experimental results.


**Audience:**

k-means is a canonical clustering algorithm, and initialization schemes have been widely studied and are important to its final success. The existence of a new scheme that can be applied on top of an existing initialization, while resulting in better performance, would appear to be of interest to a number of individuals in the TMLR audience.

**Claims And Evidence:**

The paper's main claim is that the proposed method for initializing k-means is effective at reducing the SSE cost, while introducing a modest computational overhead compared to existing methods. These claims appear to be supported, though they could be spelled out more clearly. (See detailed comments below.)